# Mortality prediction for ICU patients with mental disorders using large language models ensemble and unstructured medical notes

**Waleed Nazih[1], Tamer Abuhmed[2]\*, Meshal Alharbi[1], Shaker El-Sappagh[2,3]**

**1** Department of Computer Science, College of Computer Engineering and Sciences, Prince Sattam bin Abdulaziz University, Al-Kharj, Saudi Arabia, **2** College of Computing and Informatics, Sungkyunkwan University, Suwon, South Korea, **3** Faculty of Computer Science and Engineering, Galala University, Suez, Egypt

\* tamer@skku.edu

**Data availability statement:** MIMIC-IV clinical notes dataset is available by requesting it on https://www.physionet.org/content/mimic-iv-note/2.2/.

## Abstract

Assessing mortality risk in the intensive care unit (ICU) is crucial for improving clinical outcomes and management strategies. Conventional artificial intelligence studies often neglect vital clinical information contained in unstructured medical notes. Recently, large language models (LLMs) have achieved leading-edge performance in natural language processing tasks, though each model has limitations stemming from its architecture and pre-training. The ensemble of heterogeneous language models, including both conventional LMs and LLMs, effectively addresses these constraints. The study introduces a predictive ensemble classifier using a decision fusion approach of diverse medical LLMs and LMs, including Asclepius, Meditron, GatorTron, and PubMedBERT. These models, fine-tuned with multimodal data from the medical records of 11,914 individuals diagnosed with various mental disorders from the MIMIC-IV dataset, enhance the diversity of the resulting ensemble model. The performance of our multimodal ensemble model was rigorously evaluated, delivering superior results compared to individual LLM and LM models based on single modalities. Our study underscores the substantial influence of language models on mental health management in the ICU, advocating for advanced clinical decision-making techniques that integrate unstructured medical texts with language models to enhance patient outcomes.

## 1 Introduction

Mental health (MH) refers to an individual's emotional, psychological, and social state of well-being. It can be influenced by numerous conditions [1]. Mental conditions such as depression, anxiety, bipolar disorder, post-traumatic stress disorder, schizophrenia, anorexia nervosa, and attention deficit hyperactivity disorder are associated with significant distress and functional impairment, as well as profound disturbances in thought, emotional regulation, and behavior. According to the WHO, in 2019, one in eight people, or 970 million people worldwide,

**Funding:** The authors extend their appreciation to the King Salman Center for Disability Research for funding this work through Research Group no KSRG-2022-101.

**Competing interests:** No authors have competing interests.

experienced a mental disorder, predominantly anxiety and depression [2]. In 2020, the prevalence of patients with anxiety and depression rose markedly due to the COVID-19 pandemic (specifically, 26% and 28% increases, respectively, in one year) [3]. In 2021, the National Institutes of Health reported that 57.8 million adults aged 18 and above were experiencing some type of mental illness in the US, which constituted 22.8% of the adult population [4]. Adults aged 18-25 years exhibited a higher prevalence of 33.7% in mental disorders, which can affect functionality in all areas of life and cause problems in educational and occupational settings. In 2022, 23.1% of adults in the US encountered a mental health issue, and 6% were affected by severe disorders [5]. Individuals suffering from severe mental health disorders often have a lifespan that is 10 to 20 years shorter than that of the general population. Furthermore, mental disorders significantly affect the economy; for example, depression and anxiety result in an annual global productivity loss of $1 trillion [6]. Furthermore, the proportion of adults in the US who obtained mental health services rose from 19.2% in 2019 to 21.6% in 2021 [7].

Predicting mortality in the intensive care unit (ICU) poses a significant challenge in healthcare, as precise and prompt prognoses are essential to direct treatment decisions and improve patient outcomes [8]. This process uses extensive clinical data, including laboratory results, vital signs, medical history, and severity scores, to develop predictive models using various machine learning and statistical techniques, such as logistic regression, random forests, and neural networks. Mortality risk assessment and prediction in patients with mental disorders constitute a significant global health concern, studied to reduce preventable deaths and narrow the mortality gap between this population and the healthy population [9]. Strong evidence suggests that mental disorders significantly increase mortality rates in ICU units [9,10]. In [11], a study conducted a statistical analysis to evaluate the extent of mortality from the ICU and identify its predictors among patients in northwest Ethiopia. The study found that patients with abnormal mental status upon admission had a higher likelihood of mortality compared to those who were conscious. According to Li et al., [12], there is a notable association between mental disorders and mortality in the ICU. Vai et al., [13] investigated the link between mental disorders and the likelihood of death related to COVID-19 in ICU settings. Consequently, identifying the mortality risk in a patient with an abnormal mental state is crucial to save lives and enhance the patient's quality of life. Various machine learning and deep learning techniques have been used to predict ICU mortality. El-Rashidy et al., [8] developed a stacking ensemble model to predict ICU mortality using MIMIC-III data. Choi et al., [14] explored different machine learning models to predict ICU mortality. Chen et al., [15] introduced 1D-MSNet, a 1D multiscale convolutional neural network model to predict mortality in the ICU using the MIMIC-IV v1.0 dataset. Huang et al. [16] utilized the MIMIC-IV and eICU datasets to refine an ensemble model composed of random forest, LightGBM, and XGBoost, aimed at predicting mortality among ICU patients diagnosed with lung cancer.

NLP algorithms have been used to predict mortality in the ICU based on text notes from electronic health records [17]. NLP techniques facilitate the extraction of terms such as sepsis, fixed pupils, and coagulopathy from textual notes for use in mortality prediction models. In their study, Joshua et al. [18] combined physician notes with injury severity scores to predict mortality rates in a surgical ICU setting, and achieved an accuracy of 94.6 ± 1.1%. [19] developed a model that combined logistic regression and the TF-IDF NLP algorithm to process text notes alongside other clinical data, such as vital signs and laboratory tests, to predict ICU mortality. Mahbub et al. [20] investigated the prediction of short-, mid-, and long-term adult ICU mortality exclusively through the use of unstructured clinical notes from the MIMIC III dataset, employing various NLP methodologies such as frequency-based approaches, fixed embeddings, and dynamic embeddings. The study found that relying solely on clinical notes

from the first 24 hours of admission achieved high results. Cooley-Rieders and Zheng [21] developed an LSTM-based deep learning model to predict ICU mortality using raw nursing notes solely from the MIMIC-III dataset, achieving an ROC of 0.8629. To our knowledge, no existing studies in the literature predict ICU mortality among patients with mental disorders by analyzing clinical notes. Although these models have achieved good results, they are based on small datasets, which affects their generalizability. To the best of our knowledge, no model has yet been implemented in real clinical settings to automate the prediction of mortality, especially in intensive care units.

Recent advances in transformer-based pre-trained language models (PLM) and large language models (LLMs) have exhibited disruptive potential in various domains, including healthcare [22,23]. Their expressivity, reasoning capabilities, and ability to perform few-shot and in-context learning have enabled deeper integration into clinical workflows. Medical domain LLMs, such as Med-PaLM 2, HuatuoGPT, and Visual Med-Alpaca, have been developed to address complex challenges in diagnostics, summarization, and clinical decision support [24,25]. In mental health, LLMs have shown promise in tasks such as functional screening, psychotherapeutic intervention, and early detection of cognitive distortions [26–29]. For example, ChatGPT has been used to build educational chatbots [22,30], and models like Mental-Alpaca and Mental-FLAN-T5 have achieved superior predictive performance for mental health outcomes compared to GPT-3.5 and GPT-4 [31].

Despite these advances, the use of LM and LLM to predict ICU mortality among patients with mental disorders remains largely unexplored. Most prior work has focused on general ICU populations or specific diseases, with minimal emphasis on psychiatric comorbidities. Addressing this gap is vital, given the higher mortality risks associated with mental disorders in critical care settings. Ensemble modeling has emerged as a promising strategy for enhancing prediction performance. Techniques such as voting and boosting aggregate multiple models to improve accuracy, robustness, and generalizability [32–34]. In the context of language models, ensemble approaches can mitigate biases and limitations inherent in individual architectures by leveraging diverse pretraining sources and model configurations. For example, LLM-Blender [35] and majority-voting ensembles [33] have yielded more stable outcomes without requiring excessive computational resources. Motivated by this, we investigate the potential of ensemble-based LLMs and LMs to predict mortality in ICU patients with mental disorders, a research direction that, to the best of our knowledge, remains unexplored [36].

In this study, we propose a novel ICU mortality prediction framework for patients with mental disorders using LLMs and unstructured clinical notes. Our approach exclusively leverages discharge summaries and radiology reports to fine-tune a diverse set of LLMs and LMs. The study is guided by the following research questions, grouped into three thematic categories: RQ1: How does fine-tuning large language models on domain-specific medical datasets (e.g., MIMIC-IV) affect predictive accuracy for ICU mortality? RQ2: What are the relative contributions of discharge summaries and radiology reports in predicting ICU mortality for patients with mental disorders? RQ3: How do LLMs compare with traditional language models (LMs) in mortality prediction performance within this clinical context? RQ4: What benefits does decision fusion offer over early fusion in ensemble-based mortality prediction models? RQ5: How effectively can LLMs utilize unstructured clinical notes to enhance mortality prediction outcomes? RQ6: How does the integration of multiple modalities (i.e., discharge summaries and radiology reports) influence the predictive performance of ICU mortality models? RQ7: To what extent does ensemble modeling improve the robustness and generalizability of mortality predictions for ICU patients with mental disorders? RQ1, RQ3, RQ4, and RQ7 are related to model design and optimization, while RQ2 and RQ6 are related

to data modality and representation. Finally, RQ5 explores the applicability of unstructured clinical text. The main contributions of this study can be outlined as follows.

- We introduce a mortality prediction framework for patients with mental disorders in the ICU based on a voting ensemble model of language models (both LLM and LM) and the late fusion of multimodal data (i.e., discharge summary and radiology report). This represents the first model to predict ICU mortality for patients with mental disorders using textual data and both LLM and LM models.
- Various pre-trained LLM and LM models of different sizes from different repositories, such as HuggingFace, are employed. To manage the fine-tuning process and to create suitable models, we utilize Low-Rank Adaptation (LoRA) and model quantization techniques.
- We examine the role of different modalities independently, including discharge summary and radiology report. In addition, various modality fusion strategies, such as early fusion and decision fusion, are investigated, where different combinations of LLM and LM based on different modalities are explored.
- The ensemble model collaboration strategy is explored by combining outputs from diverse models to derive coherent and stable classification results [37].
- We explore the essential role of the fine-tuning process in expanding the knowledge of language models, which enhances the accuracy of the resulting classifiers.
- Extensive experiments were performed using the MIMIC-IV benchmark dataset, where various text modalities for patients with mental disorders, including discharge and radiology reports, were extracted. Furthermore, the fine-tuning phase is critical in developing classifiers based on LLM and LM models for specialized domains such as medicine. Furthermore, we found that ensembles of diverse LLM and LM models, utilizing heterogeneous modalities, can significantly improve the accuracy of the resultant classifiers.

The organization of this paper is structured as follows. Sect 2 reviews related research on mortality prediction in ICU settings and investigates the application of language models in the healthcare domain. Sect 3 describes the primary materials and methods, including the dataset used, the different language models (LLMs and LMs) utilized in the study, and the proposed framework for mortality prediction as a classification task. This section also details the experimental setup and the evaluation metrics used to assess model performance. Sect 4 presents and discusses the results of the experiments, compares the performance of various models, and evaluates the impact of different modalities and fusion strategies on prediction accuracy. Sect 5 addresses the limitations and future directions, while Sect 6 concludes the study's main findings.

## 2 Related work

Large language models have shown potential in predicting clinical NLP tasks [24,38,39], particularly medical tasks for patients in the ICU based on unstructured medical notes.

### 2.1 Mortality prediction in ICU using ML/DL

Evaluating the risk of mortality is complex and costly due to numerous influential factors. Physicians aim to identify ICU patients at a heightened risk for complications to address these issues efficiently. Since mortality prediction in ICUs is crucial for patient care and management, several studies have aimed to enhance the accuracy of these prediction models for patients with comorbidities. These studies have employed machine and deep learning models to predict mortality using a variety of parameters, including physiological factors, lab

results, comorbidity indices, severity of illness scores, and demographic variables [40]. [41] suggested predicting mortality in Diabetes ICU patients by converting previous hospital visits into event logs, including demographic details, diagnoses, procedures, and diabetes-specific health metrics. These logs were then processed through the Decay Replay Mining with Neural Architecture Propagation (DREAM-NAP) model, achieving an area under the receiver operating characteristic curve (AUROC) score of 0.826. The study by [42] examined how machine learning could be utilized to predict 90-day mortality among ICU patients using high-frequency data extracted from electronic patient records. The study employed a deep learning approach, using Long Short-Term Memory (LSTM) units. This model was designed to generate dynamic, real-time predictions by integrating static baseline data from ICU admission with continuous physiological data gathered during the patient's stay in the ICU. The model's initial predictions had an AUROC of 0.73 and a Matthews correlation coefficient (MCC) of 0.29 at admission, which improved to 0.85 and 0.5, respectively, after 72 hours. [8] integrated multiple machine learning algorithms based on their individual performance across different ICU data modalities to enhance mortality prediction accuracy. These classifiers are combined using a stacking ensemble approach, where predictions from the individual classifiers (level-0 models) are inputs to a second-level meta-classifier, optimizing the final prediction output. The ensemble model demonstrated high accuracy, with the best model performance achieving an accuracy of 94.4%, precision of 96.4%, recall of 91.1%, F1-score of 93.7%, and an AUROC of 93.3%. [43] employed a multilayer dynamic ensemble model to predict ICU mortality and length of stay (LoS) for neonates, using medical longitudinal data from 3,133 infants in the MIMIC dataset. The model uses a novel approach by integrating classification and regression tasks within a dynamic ensemble framework. Additionally, it incorporates explainable AI (XAI) techniques to make its decision-making process transparent, aiming to enhance trust among medical professionals [44]. The success of the ensemble model in surpassing single-model approaches demonstrates the potential of integrating multiple machine-learning methods to enhance predictive accuracy in critical healthcare settings. Li et al. [45] designed and validated a model driven by machine learning to foresee mortality in 1,177 heart failure patients admitted to the ICU, utilizing data from the MIMIC-III dataset. Extreme Gradient Boosting (XGBoost) and Least Absolute Shrinkage and Selection Operator (LASSO) regression were used to determine independent risk factors and build the predictive models. The final model achieved an AUROC of 0.8515. [46] analyzed 24 studies, including 374,365 patients, and several ML methods are discussed and evaluated, including artificial neural networks. With ML's capabilities to handle high-dimensional datasets with various variable types, there is increasing interest in incorporating ML into medical diagnosis, including mortality prediction. However, the study also highlights challenges such as issues with external study validation, results interpretability, and the potential risk of overfitting. Structured data-based clinical predictive models face limitations in routine practice because of the complicated nature of data processing [47]. However, integrating structured and unstructured electronic health record (EHR) data has proven critical for predicting ICU patient mortality [48]. [49] conducted sentiment analysis on nursing notes to explore its link with 28-day in-hospital mortality rates in sepsis patients. This method shows the potential benefits of leveraging unstructured text for mortality prediction in ICU settings. Neural language modeling pipelines have been utilized for prognostic clinical prediction using unstructured medical text notes, emphasizing the value of pre-trained language models in mortality prediction [50].

## 2.2 Mortality prediction of mental disorder patients

Several research studies have developed predictive models using electronic health records to continuously monitor and predict the risk of mental health problems [51]. A detailed and timely prediction of a broad range of mental health issues is crucial to improving patient outcomes and optimizing allocation of healthcare resources [52]. [12] examined the clinical traits and mortality risk factors in patients with mental disorders complicated by severe pneumonia. The study developed a predictive model through logistic regression analysis and created a dynamic nomogram for use in clinical settings to estimate mortality risk.

The model incorporates several critical parameters, such as age, type of primary mental illness, and specific biomarkers like procalcitonin (PCT) and hemoglobin levels, demonstrating their predictive relevance for the prognosis of these patients. The model achieved an AUC of 0.827, with a sensitivity of 73.4% and a specificity of 80.4%. [13] conducted a systematic review and meta-analysis to assess how pre-existing mental disorders affect COVID-19-related mortality. The research found that individuals with any mental disorder faced substantially higher risks of dying from COVID-19 and increased hospitalization rates, though this did not include ICU admissions. Specific mental disorders such as psychotic, mood, and substance use disorders were notably linked with higher mortality risks. Additionally, exposure to psychopharmacological drugs like antipsychotics and anxiolytics significantly raised mortality risk. These results underscore the need for increased focus and customized interventions for individuals with mental disorders to reduce their heightened risks of COVID-19-related mortality. [53] corroborated the previous study's findings on the mortality risk among patients with serious mental illnesses (SMI) hospitalized for COVID-19, considering various confounding factors such as medical comorbidities, admission clinical status, and treatment modalities. The findings show that patients with SMI experienced a significantly higher in-hospital mortality rate (11.17%) compared to non-SMI patients (3.27%). [54] explored the relationship between Posttraumatic Stress Disorder (PTSD) and mortality, based on data from over 2.1 million participants diagnosed with or exhibiting symptoms of PTSD. Their analysis from this systematic review and meta-analysis indicated that PTSD is linked with a significantly elevated mortality risk. Specifically, individuals with PTSD exhibit a 47% higher mortality risk across studies that report Odds Ratio and Risk Ratio, and a 32% increased risk in studies using mortality analyses. Notably, this increased mortality risk is more pronounced among civilian populations compared to military veterans. The study also underscores the importance of further research, particularly on civilian populations, women, and individuals from less developed countries, to fully comprehend the long-term effects of PTSD on mortality. [10] systematically explored studies to identify the relationship between various mental disorders and suicide risk. Employing meta-regression analysis, the study evaluated data from multiple studies to provide adjusted relative risks (RRs) for suicide linked to specific mental disorders such as major depressive disorder, dysthymia, bipolar disorder, schizophrenia, and anxiety disorders. The analysis highlights that all examined disorders significantly predict increased suicide mortality risk, particularly emphasizing the high risks associated with mood disorders and schizophrenia. Deep learning has demonstrated promising results in predicting mortality for patients with mental disorders by using clinical data and advanced DL models. Using a dataset of 20,000 US veterans diagnosed with bipolar disorder, Shao Y. et al., [55] extracted features from a variety of clinical data points recorded up to one year before and immediately after diagnosis. In their work, they transformed these data into temporal images, which were then utilized to train DL models. The models achieved prediction accuracies ranging from 76.6% to 94.9% and AUC from 0.745 to 0.806 for combined hospitalization and mortality outcomes. [56] introduced a method to enhance the interpretability

of machine learning predictions in the context of SMI, such as schizophrenia. The research used class-contrastive counterfactual reasoning to show how modifications to specific inputs can alter mortality predictions using data from a secondary mental health care provider. The developed model successfully predicted mortality with an AUROC of 0.80. Factors such as the use of antidepressants and substance abuse were identified as significantly impacting mortality risks.

The research literature lacks studies that thoroughly assess the effectiveness of LLM in constructing models to predict mortality, despite growing interest. In the healthcare field, the use of LLM primarily focuses on tasks such as medical text summarization [24], generating discharge summaries, extracting clinical concepts, answering medical questions, interpreting electronic health records, and generating medical articles, rather than on clinical prediction [57]. To our knowledge, this is the first study to explore the utilization of LMs/LLMs in predicting mortality among patients with mental disorders.

## 3 Materials and methods

In this section, we present the materials and methods used in our study to predict ICU mortality for patients with mental disorders using advanced language models. We begin by the dataset details, extracted from the MIMIC-IV clinical notes, including discharge summaries and radiology reports. Following this, we explore the various pre-trained language models (LLMs and LMs) used in the study and the fine-tuning processes applied to them. We then describe the proposed framework for predicting patient mortality, highlighting the decision fusion approach that combines multiple model outputs for accurate and stable decisions. Finally, we describe the evaluation metrics and experimental setup of how the models were trained, fine-tuned, and assessed.

### 3.1 Dataset

This study introduces a framework that uses language models to assess mortality risks in patients with mental disorders. We employ the MIMIC-IV clinical notes dataset [58]. It contains anonymized clinical notes from ICU patients, including 331,794 discharge summaries from 145,915 patients and 2,321,355 radiology reports from 237,427 patients. In compliance with the Health Insurance Portability and Accountability Act (HIPAA), sensitive and protected health information has been removed by dataset owners but is securely linked to other patient data, providing crucial context for domain experts.

We extracted essential patient data, including anonymized death dates, from the MIMIC-IV dataset [59]. We obtained patient discharge summaries and radiology reports from the MIMIC-IV-Note: Deidentified free-text clinical notes dataset [58]. To ascertain whether deceased patients died in or out of the hospital, we used the "deathtime" field from the "admissions" table and the "dod" (date of death) field from the "patients" table. Patients who died during their first admission are excluded from this group. We identified patients with mental disorders using ICD-10, which offers higher specificity and flexibility, thus enhancing the accuracy and comprehensive support of healthcare data. The mental disorder codes in ICD-10 start with 'F', such as "F419" which refers to anxiety disorder. Fig 1 shows the frequency of ICD-10 codes for the mental disorders that exist more than twenty times in the dataset. The most frequent code was "F329", which stands for unspecified major depressive disorder. Schizophrenia, depression, and bipolar disorder have large distributions in the used dataset, i.e., F200: Paranoid schizophrenia (28 patients), F209: Schizophrenia (57 patients), F319: Bipolar disorder, unspecified (218 patients), and F329: Major depressive disorder (2,394 patients). In addition, some codes rarely appear in the dataset, such as "F508" which refers to

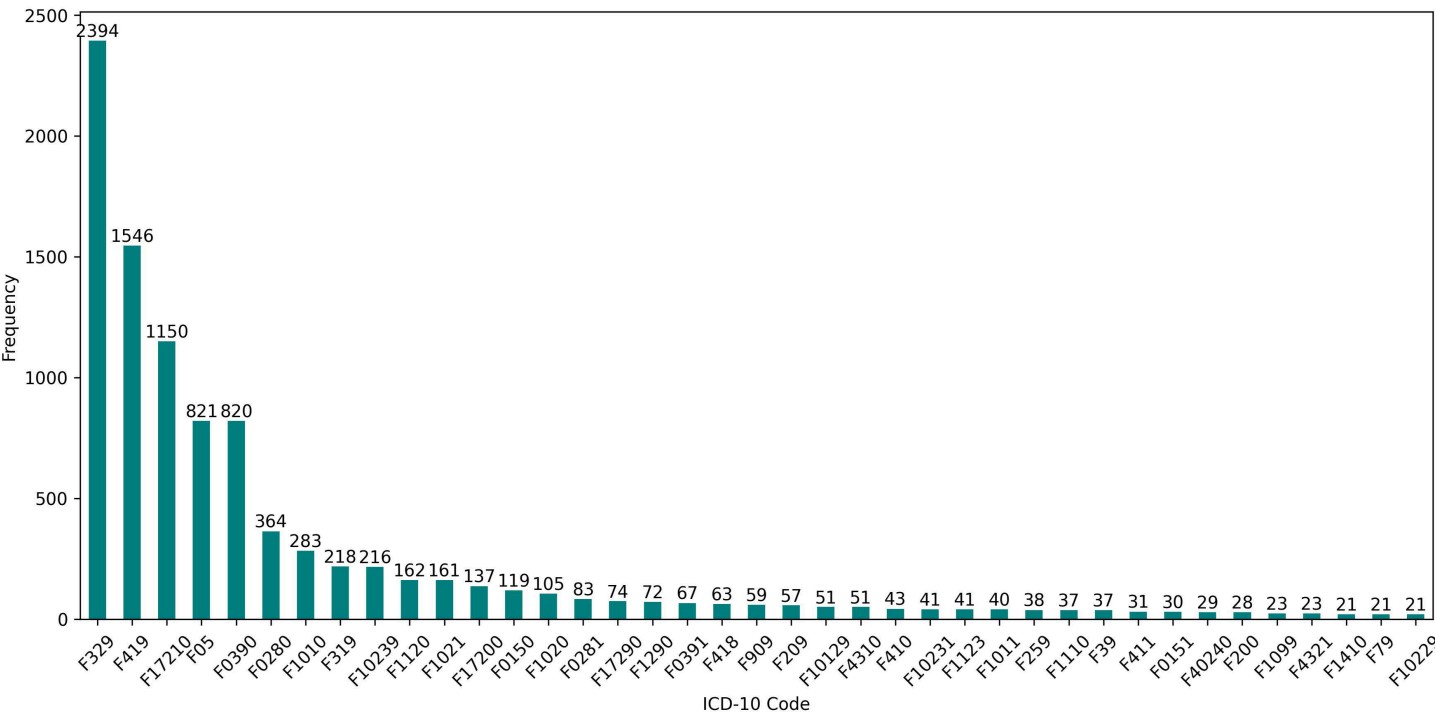

**Fig 1. Frequency of mental disorders ICD-10 codes in the dataset.**

other eating disorders, and "F901," which stands for the predominantly hyperactive-impulsive type of attention-deficit hyperactivity disorder.

From the dataset, we identified 5,957 deceased ICU patients with mental disorders. As our study focuses on this population, all included subjects were diagnosed with mental disorders. For evaluation, we filtered 29,419 such patients, among whom 4,684 died in the ICU and 1,273 died post-discharge, totaling 5,957 deceased cases. To balance the classification task, we randomly sampled 5,957 survivors from the remaining patients, resulting in 11,914 patients with discharge summaries (5,844 [49%] males). Discharge summaries range from 127 to 7,354 words, with an average of 1,790±740 words. Among these, 10,243 patients (5,108 [50%] males) also have radiology reports, including 5,121 deceased cases. The exclusion of 1,671 subjects without radiology reports reduced the sample size available for fusion-based model training. Radiology reports range from 21 to 22,005 words, with an average length of 983±1,273 words. Fig 2 shows the distribution of word counts in the discharge summaries and radiology reports, respectively.

### 3.2 Language models

Pretrained language models have shown high effectiveness in managing natural language processing tasks, especially with unstructured text. This study examines pre-trained language models listed in Table 1 that are initially trained on medical data. We explore the capabilities of these models, including PubMedBERT, Clinical BERT, and RoBERTa-large, CORe-clinical-outcome-bioBERT, Clinical-Longformer, Clinical-Bird, and Gatortron-Base, as well as large language models like Asclepius 7B and Meditron 7B, GatorTron-Medium 3.9B, and Gatortron-Large 8.9B. For this study, a model with at least 1B parameters is considered an

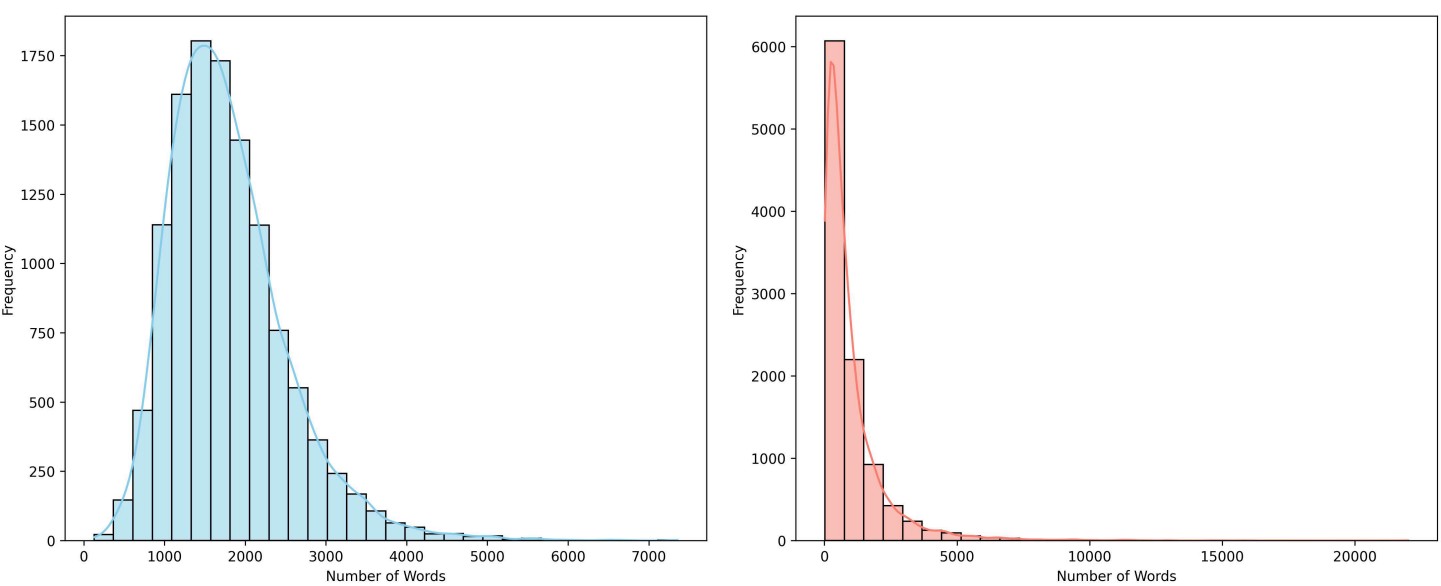

**Fig 2. Frequency of word counts in dataset.** Left sub-figure: discharge summaries and right sub-figure: radiology reports.

**Table 1**. The utilized LLM and LM models.

| Model | Parameters # | # Train Parameters | Domain | Sample Size | Sequence Length | Data Source | Perplexity Score | Transformer | Architecture |
|---|---|---|---|---|---|---|---|---|---|
| Asclepius [63] | 7B | 16.7M | Clinical Notes | 57k | 2048 | MIMIC-III | 2.186 | Y | LlamaForCausalLM |
| Asclepius-R [71] | 7B | 16.7M | Clinical Notes | 57k | 2048 | MIMIC-V | 2.809 | Y | LlamaForCausalLM |
| Meditron [72] | 7B | 16.7M | Clinical Notes | 46k | 2048 | MIMIC-V | 2.809 | Y | LlamaForCausalLM |
| Gatortron-Large [70] | 8.9B | 19.2M | Clinical Notes | 90B | 512 | MIMIC-III,PubMed | 2.809 | Y | LlamaForCausalLM |
| GatorTron-Medium [70] | 3.9B | 11.8M | Clinical Notes | 90B | 512 | MIMIC-III,PubMed | 2.809 | Y | LlamaForCausalLM |
| BiomedBERT-A [64] | 110M | 591K | PubMed Abstracts | 3.1B | 512 | PubMed | 3.32 | X | BERT |
| BiomedBERT-F [64] | 340M | 591K | PubMed Full-text | 16.8B | 512 | PubMed | 3.54 | X | BERT |
| Clinical BERT [73] | 137M | 1.9M | Clinical Notes | 1.2B | 256 | MIMIC-III | 8.67 | X | BERT |
| BioClinicalBERT [65] | 110M | 886K | Clinical Notes | 880M | 128 | MIMIC-III | 4.48 | X | BERT |
| Bio Discharge Summary BERT [65] | 110M | 886K | Discharge Summaries | 880M | 128 | MIMIC-III | NA | X | BERT |
| RoBERTa-Large [66] | 355M | 3.4M | Text | 160GB | 512 | Five text datasets | 1378.71 | Y | BERT |
| RoBERTa-Base [66] | 125M | 1.47M | Text | 160GB | 512 | Five text datasets | 1378.71 | Y | BERT |
| CORe-clinical [67] | 109M | 591K | Admission Notes | NA | 512 | MIMIC-III,PubMed | NA | Y | BioBERT |
| Gatortron-Base [70] | 357M | 2.3M | Clinical Notes | 90B | 512 | MIMIC-III,PubMed | NA | Y | LlamaForCausalLM |
| Clinical-Longformer [69] | 150M | 1.4M | Clinical Notes | 16.5K | 4096 | MIMIC-III | 1.61 | Y | Longformer |
| Clinical-Bird [68] | 130M | 1.4M | Clinical Notes | 16.5K | 4096 | MIMIC-III | 1.41 | Y | BigBird |

LLM; otherwise, it is considered an LM [60,61]. As models have grown larger, this threshold provides a practical distinction that reflects observed differences in behavior, capability, and resource requirements [62]. Specifically, models with over 1B parameters typically demonstrate enhanced generalization, emergent reasoning abilities, and higher computational demands, characteristics broadly associated with LLMs.

The LLMs examined use a transformer backbone architecture that includes self-attention networks (SANs) and feedforward networks (FFNs). SANs process the relationships among tokens in the input sequence, whereas FFNs focus predominantly on enhancing the inner nonlinear transformations of the tokens. Each transformer layer comprises both a SAN module and an FFN module. The input $X$, representing the word embedding for each token combined with positional embeddings, is provided to the SAN, where it is linearly projected into query, key, value, and output spaces $\{Q, K, V\}$ as described below:

$$\begin{bmatrix} Q \\ K \\ V \end{bmatrix} = X \times \begin{bmatrix} W^Q \\ W^K \\ W^V \end{bmatrix} \tag{1}$$

The self-attention mechanism based on the query, key, and value is calculated as:

$$Attention(Q, K, V) = softmax\left(\frac{QK}{\sqrt{d_k}}\right) V \tag{2}$$

As most transformers use a multi-head variant of the self-attention mechanism, they execute the self-attention mechanism $h$ times, applying linear projections of Q, K, and V to reduced dimensions (e.g., $d_k = \frac{d_{model}}{h}$) instead of utilizing a single attention function with $d_{model}$-dimensional keys, values, and queries. Finally, the output of SAN is

$$SAN(X) = [head_1; \cdots; head_h] W^O, s.t. head_i = Attention(Q_i, K_i, V_i) \tag{3}$$

where individual attention heads are independently calculated as $Q = [Q_1; \cdots Q_h]$, $K = [K_1; \cdots K_h]$, and $V = [V_1; \cdots V_h]$. A non-linear activation-based feed-forward network is alternately employed with every SAN layer as:

$$FFN(X) = \delta(XW^{in}) W^{out} \tag{4}$$

The function $FFN(X)$ employs various optimization techniques, including layer normalization, residual connections, dropout, and weight loss, to prevent overfitting problems in deep network structures.

Asclepius [63] is a clinical language model based on Llama2, trained on synthetic clinical notes from publicly accessible case reports and biomedical literature, further fine-tuned using clinical notes from the MIMIC-III dataset. BiomedBERT abstract and BiomedBERT full [64] are developed on the BERT architecture, trained from scratch with PubMed abstracts and PubMedCentral full text articles, respectively. [65] developed two clinical LMs named ClinicalBERT and Bio ClinicalBERT. The former relies on BERT embeddings, while the latter uses BioBERT embeddings [56]. Both are fine-tuned on the entirety of MIMIC III discharge notes, encompassing all entries from the NOTEEVENTS table ($\approx$ 880M words). RoBERTa-large-PM-M3 [66] is a clinical RoBERTa language model, which stands for Robustly Optimized BERT Pre-training Approach. It enhances BERT by adjusting critical hyperparameters, omitting the next-sentence prediction task, and using significantly larger mini-batches and higher

learning rates. Clinical RoBERTa is pre-trained on data from PubMed, PMC, and MIMIC-III clinical notes. CORe-clinical, also known as CORe (Clinical Outcome Representations), was introduced by [67]. It is derived from BioBERT and further trained on clinical notes, disease descriptions, and medical articles, utilizing a specialized Clinical Outcome Pre-Training objective.

Clinical-Bird [68] and Clinical-Longformer [69] are fine-tuned BigBird models that have been pretrained on MIMIC-III clinical notes. BigBird is a transformer that utilizes sparse attention to enhance models like BERT to process significantly longer sequences. Incorporating both global and random attention mechanisms into the input sequence greatly improves the computational efficiency for extended sequences. In various NLP tasks involving lengthy documents, such as question-answering and summarization, BigBird has demonstrated superior performance compared to BERT and RoBERTa. [70] introduced Gatortron-Base, developed collaboratively by the University of Florida and NVIDIA. This clinical language model, comprising 345 million parameters, uses a BERT architecture and was pre-trained on 82 billion words from clinical notes at the University of Florida Health System, 6.1 billion words from PubMed, 2.5 billion words from WikiText, and 0.5 billion words from MIMIC III clinical notes. All LLM and LM that are based on Llama utilize a decoder structure, whereas BERT-based models feature encoder-based structures.

## 3.3 Proposed framework

Fig 3 illustrates the entire proposed framework of our study. The clinical notes from the discharge summary and the radiology report are extracted from the MIMIC-IV clinical notes dataset. We begin by pre-processing the unstructured text data. Based on previous findings, we hypothesize that excessive pre-processing during pre-training can hinder generalization and unnecessarily prolong training time [38]. Therefore, we apply minimal preprocessing: redacted details such as admission and discharge dates are removed, as well as repeated newlines, underscores, non-alphanumeric, and non-punctuation characters. The text is converted to lowercase, and the extra whitespace and the equal signs are removed [20,38]. Tokenization was performed using the native tokenizer of each model, for example, WordPiece for BERT-based models and SentencePiece for LLaMA-based models, to ensure compatibility with respective embedding schemes and vocabulary coverage. Each tokenized sequence was truncated or padded to match the model's maximum input length, with padding applied post-sequence using special tokens where applicable. We formulate the task as a binary classification, using a combined input of each patient's discharge summary and radiology reports to predict survival or death. The dataset is divided into training/validation sets (90%) and test sets (10%). Each model uses its own tokenizer to process inputs, which are tokenized and padded to a fixed length. All selected models are pretrained on clinical data from various repositories, each employing a unique embedding strategy. Models are fine-tuned based on size: those with more than 1B parameters are quantized and then fine-tuned using LoRA; models under 1B are directly fine-tuned using LoRA. LoRA introduces trainable low-rank decomposition matrices into each layer of the transformer architecture, enabling task-specific adaptation without modifying the full weight matrices. This approach is particularly advantageous for LLM, where full fine-tuning can be computationally intensive and memory-inefficient [74]. Compared to conventional fine-tuning approaches, such as full parameter updates or adapter-based methods, LoRA significantly reduces the number of trainable parameters while maintaining or even improving downstream performance. This parameter efficiency is critical in our study, which involves fine-tuning a heterogeneous ensemble of LMs and LLMs with varying architectures and sizes. Moreover, LoRA has demonstrated

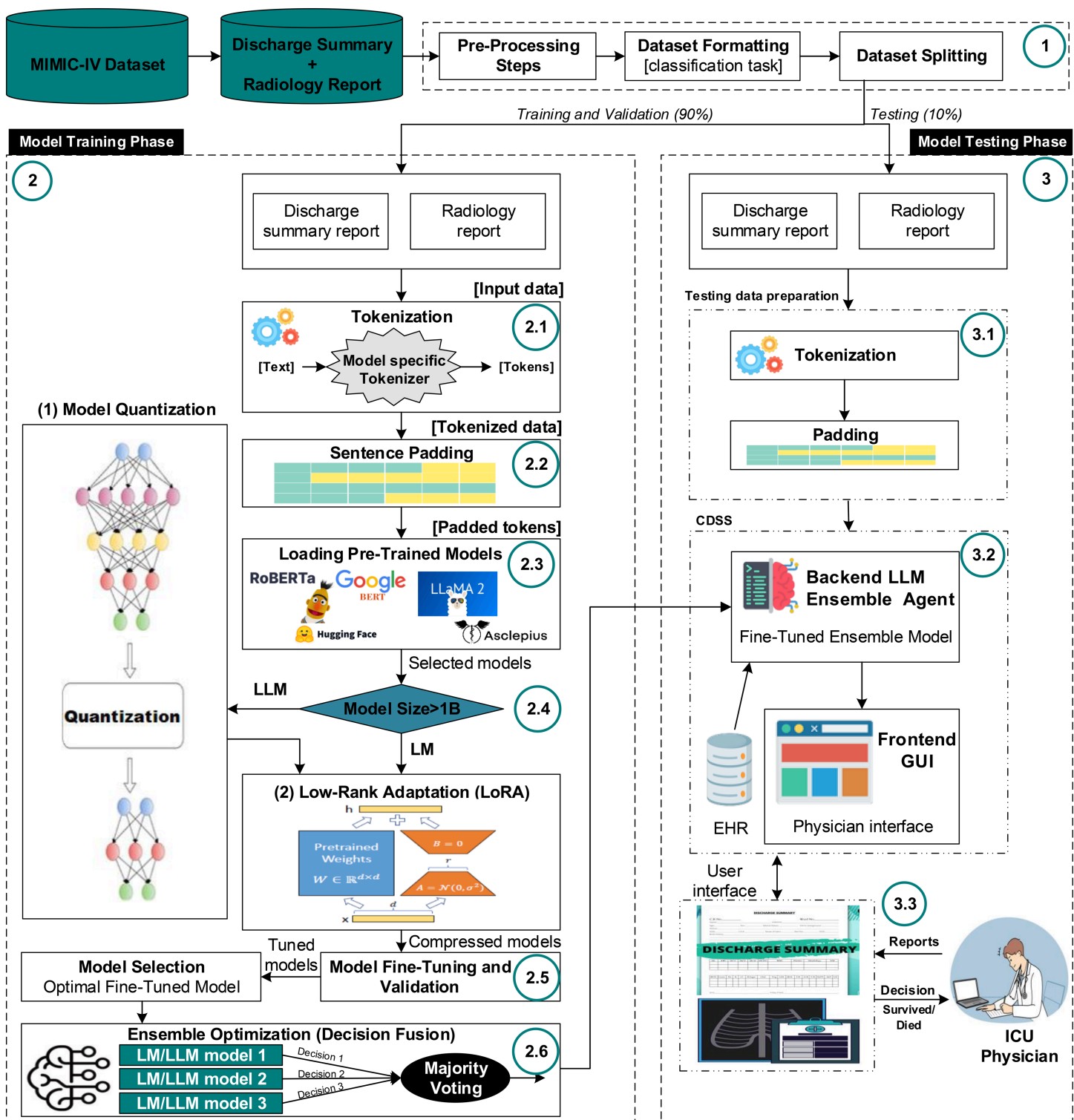

**Fig 3. Proposed LLM-based mortality prediction framework for mental-disordered patients in ICU.**

strong empirical performance in previous medical NLP applications, making it well-suited for domain-specific tasks involving unstructured clinical text. By adopting LoRA, we were able to uniformly and efficiently fine-tune models of different scales, particularly those exceeding one billion parameters, while minimizing computational overhead and facilitating reproducibility. A redrawn illustration of the LoRA mechanism, adapted for our specific training setup, is provided in Fig 4. The original concept is based on Hu et al. [74]. The used uniform quantization format is as follows [75]:

$$X_{INT} = \frac{X_{FP16} - Z}{S}, \tag{5}$$

$$S = \frac{max\left(F_{FP16}\right) - min\left(F_{FP16}\right)}{2^{N-1} - 1} \tag{6}$$

where $X_{FP16}$ represents the 16-bit floating point number (FP16), and $X_{INT}$ represents the low-precision integer number. $N$ indicates the number of bits. $S and Z$ denote the scaling factor and the zero point, respectively. In the case of symmetric quantization, the zero point $Z$ is set to zero. For asymmetric quantization, $Z equals min\left(F_{FP16}\right)$. The fine-tuning has been conducted employing LoRA. As shown in Fig 4, rather than updating all parameters during fine-tuning, LoRA focuses on updating a smaller subset of parameters, leading to significant savings in computational resources and memory. Using LoRA for fine-tuning results in weights that are very compact, facilitating easier storage and sharing. More formally, for a pre-trained weight matrix $W_0 \in R^{d \times k}$, LoRA limits its update by expressing it through a decomposition of low rank:

$$W_0 + \Delta W = W_0 + BA \tag{7}$$

where $B \in R^{d \times r}$, $A \in R^{r \times k}$ and rank $r \gg min(d, k)$. In the training process, $W_0$ is frozen and no gradient updates are done, but A and B contain trainable parameters. $W_0$ and $\Delta W = AB$ receive identical input, and their resulting vectors are added together component by component. For $h = W_0 x$, its modified forward pass is $h = W_0 x + \Delta W x = W_0 x + ABx$. B is initialized with zero, and A is initialized with random Gaussian. As a result, $\Delta W = AB$ is zero in

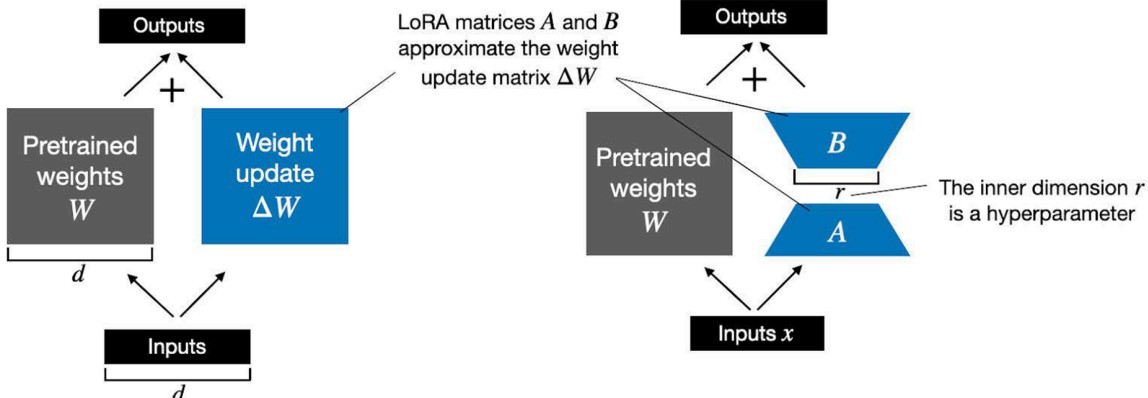

**Fig 4. LoRA re-parametrization: only the low-rank matrices A and B are trained, while the original weight matrix $W_0$ remains frozen.** The update $\Delta W = B \cdot A$ serves as a low-rank approximation to full fine-tuning, significantly reducing the number of trainable parameters.

the initial training step. Then, $\Delta Wx$ is scaled by $\alpha r$, where $\alpha$ is a constant in $r$. This process is illustrated in Fig 4.

LLM and LM models are fine-tuned and validated using the training dataset, then passed into the model testing phase. The test dataset is prepared using the same pipeline as the training dataset, consisting of tokenization, padding, and embedding. Models are tested in the binary classification task of predicting mortality in mentally disordered ICU patients. Ensemble models are established to boost the performance of their baseline classifiers. Several techniques for model ensembling exist, such as stacking and boosting. A commonly used integration technique is voting, which does not require retraining of base models, making it suitable for large models like ours. The voting technique relies on decision fusion from a list of base models. Although fusion can utilize various methods, majority voting is the most recognized. Majority voting is a widely used and effective ensemble technique due to its simplicity and robustness, especially when the individual models in the ensemble have comparable performance and are diverse in nature [76–78]. Given these properties and to maintain a focused experimental scope, we selected majority voting as a representative fusion method.

Within our framework, after tuning the baseline LLM and LM models, we explore the effectiveness of voting in enhancing mortality prediction performance, investigating different combinations of LLM and LM models with varying input data. It was found that the ensemble models significantly improve performance, outperforming all baseline LLM and LM models. The most effective model has been selected as the inference agent for a clinical decision support system in the ICU, which assists physicians in predicting patient mortality using discharge notes, radiology reports, or both. Patient's medical data can also be automatically extracted from the electronic health record, providing timely decisions.

## 3.4 Evaluation metrics

Four widely used metrics were used to evaluate the effectiveness of LLM / LM-based classifiers: accuracy, precision, recall, and F1 score. Here, TP represents the count of true positives, TN represents the count of true negatives, FP represents the count of false positives, and FN represents the count of false negatives (see Equations 8–11).

$$Accuracy = \frac{TP + TN}{TP + FP + TN + FN}. \tag{8}$$

$$Precision = \frac{TP}{TP + FP} \tag{9}$$

$$Recall = \frac{TP}{TP + FN} \tag{10}$$

$$F1 = \frac{2 \cdot precision \cdot recall}{precision + recall} \tag{11}$$

## 3.5 Experimental setup

For LLM model tuning, a server equipped with two AMD 64-Core Processors, 256 GB of memory, and a NVIDIA A40 GPU with 48GB of GPU memory was utilized. The development environment included Python 3.10.14, Hugging Face Transformers Version 4.31.0, and Hugging Face Parameter-Efficient Fine-Tuning (PEFT) Version 0.10.1.dev0. For LM tuning,

a second server was used, featuring a CPU Intel Core i7-12700F 2.1 GHz 12-Core, 64GB of memory, an SSD M.2 Kingston NV2 1TB, and a GPU ZOTAC GEFORCE RTX 4080 SUPER TRINITY 16GB. Python 3.10.12, Hugging Face Transformers Version 4.31.0, and Hugging Face PEFT Version 0.10.1.dev0. All experiments used the following experimental settings: a batch size of 4, a learning rate of $1E-4$, and the Adam optimizer. We tuned the model for six epochs for the LLM, while the LM models were tuned for 50 epochs with early stopping at 10 epochs.

## 4 Results and discussion

In this section, we investigate the performance of various language models across different sizes and sequence lengths. We evaluated the performance of different modalities, including discharge summaries and radiological reports, to predict the mortality of patients with mental disorders. Optimizing language models with additional patient data is anticipated to enhance the model performance. We explore various fusion mechanisms and present the results. The statistical significance of the models evaluated for each experiment is determined using the analysis of variance (ANOVA) test with $\alpha = 0.05$.

### 4.1 Performance of LLM and LM models without fine-tuning

This experiment investigates the research question RQ1 as fine-tuning is an essential step in updating the knowledge of language models, irrespective of size. This step is particularly crucial in the medical domain, enabling the model to make medically relevant decisions. This experiment establishes the baseline performance of the LLM and LM models and underscores the necessity of refining the general knowledge of these models with domain-specific data. It is apparent that the models perform poorly compared to their fine-tuned counterparts, which are discussed later in this section. Table 2 displays the best results of the LLM and LM models tested before fine-tuning. The top-performing LLM model, Asclepius, at a sequence length of 2048, achieved results of 0.5520 for F1-Score, 0.4576 for Accuracy, 0.4704 for Precision, and 0.6678 for Recall. The other LLM models showed even poorer results, some performing worse than random guesses. Gatortron-Base and Clinical-Bird, with 512 sequence lengths, registered the best LM results of 0.6670 for F1-Score, 0.5004 for Accuracy, 0.5004 for Precision, and 1.0000 for Recall. It is evident that, without critical fine-tuning, language models are unsuitable for specific domains such as medicine. Fine-tuning optimizes model performance and updates the model's parameters, making it more knowledgeable and appropriate for the domain. This shows the significance of fine-tuning language models on specific medical datasets, such as MIMIC-IV, influencing predictive precision for ICU mortality. In subsequent experiments, we report results after the fine-tuning process.

### 4.2 Results based on single modality: Discharge summaries

This experiment investigates the research questions RQ2 and RQ3. These questions focus on how different types of medical data, specifically discharge summaries, contribute to the prediction of mortality in the ICU in mentally disordered patients and to evaluate the comparative performance of LLM and traditional LM to make these predictions. Our goal is to explore the distinct value of discharge summaries data and assess the performance of fine-tuned LLM/LM models in a crucial healthcare setting application. Table 3 displays the test results for various LLM and LM models fine-tuned with patient discharge summary reports. Initially, it is evident that all fine-tuned models significantly outperform the untuned versions. As the table indicates, one can compare different versions of the same model on the basis of

**Table 2. Results of LLM and LM based on discharge summaries before fine-tuning.**

| Ref. | Model | Sequence Length | F1-Score | Accuracy | Precision | Recall |
|------|-------|-----------------|----------|----------|-----------|--------|
| [63] | Asclepius | 2048 | 0.5520 | 0.4576 | 0.4704 | 0.6678 |
| [72] | Meditron | 2048 | 0.3738 | 0.5021 | 0.5043 | 0.2970 |
| [70] | Gatortron-Large | 512 | 0.2817 | 0.5290 | 0.5946 | 0.1846 |
| [70] | GatorTron-Medium | 512 | 0.2924 | 0.5164 | 0.5459 | 0.1997 |
| [65] | Bio Discharge Summary BERT | 512 | 0.6548 | 0.4971 | 0.4987 | 0.9530 |
| [70] | Gatortron-Base | 512 | 0.6670 | 0.5004 | 0.5004 | 1.0000 |
| [69] | Clinical-Longformer | 512 | 0.6534 | 0.5021 | 0.5013 | 0.9379 |
| [68] | Clinical-Bird | 512 | 0.6670 | 0.5004 | 0.5004 | 1.0000 |

**Table 3. The results of LLM and LM are based on the discharge summaries.**

| Ref. | Model | Sequence Length | F1-Score | Accuracy | Precision | Recall |
|------|-------|-----------------|----------|----------|-----------|--------|
| [63] | Asclepius | 512 | 0.7470 | 0.7531 | 0.7668 | 0.7282 |
| [63] | Asclepius | 1024 | 0.7441 | 0.7557 | 0.7819 | 0.7097 |
| [63] | Asclepius | 2048 | 0.7884 | 0.7800 | 0.7601 | 0.8188 |
| [71] | Asclepius-R | 512 | 0.7130 | 0.6675 | 0.6276 | 0.8255 |
| [71] | Asclepius-R | 1024 | 0.7061 | 0.6910 | 0.6738 | 0.7416 |
| [71] | Asclepius-R | 2048 | 0.7651 | 0.7557 | 0.7372 | 0.7953 |
| [72] | Meditron | 512 | 0.7209 | 0.7380 | 0.7720 | 0.6762 |
| [72] | Meditron | 1024 | 0.7500 | 0.7615 | 0.7889 | 0.7148 |
| [72] | Meditron | 2048 | 0.7666 | 0.7674 | 0.7699 | 0.7634 |
| [70] | Gatortron-Large | 512 | 0.7866 | 0.7649 | 0.7207 | 0.8658 |
| [70] | GatorTron-Medium | 512 | **0.7982** | **0.7733** | **0.7197** | **0.8960** |
| [64] | BiomedBERT-abstract | 512 | 0.7258 | 0.6961 | 0.6616 | 0.8037 |
| [64] | BiomedBERT-full | 512 | 0.7129 | 0.6700 | 0.6313 | 0.8188 |
| [73] | Clinical BERT | 512 | 0.7278 | 0.6784 | 0.6313 | 0.8591 |
| [65] | Bio ClinicalBERT | 512 | 0.6975 | 0.6432 | 0.6057 | 0.8221 |
| [65] | Bio Discharge Summary BERT | 512 | 0.7334 | 0.6801 | 0.6291 | 0.8792 |
| [66] | RoBERTa-Large | 512 | 0.7823 | 0.7565 | 0.7079 | 0.8742 |
| [66] | RoBERTa-Base | 512 | 0.7614 | 0.7221 | 0.6675 | 0.8859 |
| [67] | CORe-clinical | 512 | 0.7278 | 0.6885 | 0.6467 | 0.8322 |
| [70] | Gatortron-Base | 512 | 0.7868 | 0.7607 | 0.7099 | 0.8826 |
| [69] | Clinical-Longformer | 512 | 0.7446 | 0.7229 | 0.6911 | 0.8070 |
| [69] | Clinical-Longformer | 1024 | 0.7728 | 0.7641 | 0.7457 | 0.8020 |
| [69] | Clinical-Longformer | 2048 | **0.7970** | **0.7725** | **0.7199** | **0.8926** |
| [68] | Clinical-Bird | 512 | 0.7490 | 0.7305 | 0.7013 | 0.8037 |
| [68] | Clinical-Bird | 1024 | 0.7562 | 0.7439 | 0.7221 | 0.7936 |
| [68] | Clinical-Bird | 2048 | 0.7852 | 0.7716 | 0.7418 | 0.8339 |

size and sequence length. Among the LLM models, three versions of Asclepius with varying sequence lengths are analyzed. The model with the longest sequence length (2,048) shows superior performance, achieving F1 scores, precision rates, and recall values of 0.7884, 0.7800, 0.7601, and 0.8188, respectively. Asclepius also produces comparable results, with a consistent F1 score of 0.74, using sequence lengths of 512 and 1024. Three Asclepius-R models with sequence lengths of 512, 1024, and 2048 were fine-tuned. The 2048 sequence length demonstrated superior performance, achieving F1 scores, accuracy, precision, and recall of 0.7651, 0.7557, 0.7372, and 0.7953, respectively. A similar trend was observed in the Meditron model,

where the 2048 sequence length model also performed best. For GatorTron, the medium-sized model (GatorTron-Medium) with a 512-sequence length outperformed GatorTron-Large with identical sequence length, registering 0.7982, 0.7733, 0.7197, and 0.8960 for F1 score, accuracy, precision, and recall, respectively. The superior results of the GatorTron-Medium model may be attributed to the size of the fine-tuning dataset, which is adequate for the medium model compared to the larger one. Upon comparison, the GatorTron-Medium with 512 sequence length consistently delivered the best results among all evaluated LLM models ($P < 0.05$).

Regarding LM models, various models of different sizes have been fine-tuned and tested. The test results are displayed in Table 3. Both BiomedBERT Abstract and Full were evaluated; BiomedBERT Abstract with a 512 sequence length demonstrated superior performance (namely, 0.7258, 0.6961, 0.6616, 0.8037 for F1 score, accuracy, precision, and recall, respectively). The ClinicalBERT model exhibited better performance than its expanded counterpart, BioClinicalBERT (i.e., 0.7278, 0.6784, 0.6313, and 0.8591 for F1-score, accuracy, precision, and recall, respectively), where both models are tuned and tested using a 512 sequence length. Surprisingly, the Bio-Discharge-Summary-BERT with a 512 sequence length does not achieve satisfactory results compared to other language models, despite being fine-tuned with discharge summaries. With a sequence length of 512, RoBERTa-Large and RoBERTa-Base are fine-tuned and tested. The test results indicate that RoBERTa-Large outperforms the base version, achieving 0.7823, 0.7565, 0.7079, and 0.8742 for F1 score, accuracy, precision, and recall, respectively. The Clinical-Longformer model is fine-tuned using three sequence lengths (i.e., 512, 1024, and 2048). We observe that the model with the longest sequence exhibits superior performance compared to the other two LM models. It achieves results of 0.7970, 0.7725, 0.7199, and 0.8926 for the F1-score, accuracy, precision, and recall, respectively. The Clinical-Bird model, evaluated with the same sequence lengths, also demonstrates that the model with the longest sequence achieves the best results compared to the other two ($P < 0.05$), specifically 0.7852, 0.7716, 0.7418, and 0.8339 for the F1 score, accuracy, precision, and recall, respectively. However, CORe-clinical shows lower performance relative to all other LMs (i.e., 0.7278, 0.6858, 0.6467, and 0.8322 for F1 score, accuracy, precision, and recall, respectively), but Gatortron-Base achieves superior results, recording 0.7868, 0.7607, 0.7099, and 0.8826 for F1-score, accuracy, precision, and recall are comparable with the best LM model, namely the Clinical-Longformer. The best LLM is GatorTron-Medium, while the best LM remains the Clinical-Longformer. Although LLM models generally outperform LM models on average, there is no significant difference in performance between the top LLM and the top LM. From this experiment, we show a comparative performance evaluation of fine-tuned LLM and LM pre-trained models in predicting ICU mortality among patients with mental disorders.

## 4.3 Results based on single modality: Radiology reports

This experiment addresses research questions RQ2 and RQ3, focusing on the role of radiology reports in predicting ICU mortality among mentally disordered patients. It also aims to compare the performance of LLMs and traditional LMs in making these predictions. The goal is to understand the distinct contributions of radiology report data and assess the effectiveness of fine-tuned LLM/LM models for this task. Table 4 displays the testing results of various LLM and LM models, calibrated with patients' radiology reports. The same conclusion was also observed in this experiment. All fine-tuned models based on radiology reports yield superior results compared to untuned models. As demonstrated in the table, various versions of the same model, differing in size and sequence length, can be compared. In the case of Asclepius LLM models with varying sequence lengths, the version with the highest length,

Table 4. The results of LLM and LM are based on the radiology reports.

| Ref. | Model | Sequence Length | F1-Score | Accuracy | Precision | Recall |
|---|---|---|---|---|---|---|
| [63] | Asclepius | 512 | 0.7062 | 0.6767 | 0.6797 | 0.7348 |
| [63] | Asclepius | 1024 | 0.7398 | 0.6952 | 0.6742 | 0.8195 |
| [63] | Asclepius | 2048 | 0.7415 | 0.7176 | 0.7185 | 0.7661 |
| [71] | Asclepius-R | 512 | 0.7645 | 0.7186 | 0.6857 | 0.8637 |
| [71] | Asclepius-R | 1024 | 0.7321 | 0.7371 | 0.7935 | 0.6796 |
| [71] | Asclepius-R | 2048 | 0.7643 | 0.7352 | 0.7218 | 0.8122 |
| [72] | Meditron | 512 | 0.6718 | 0.6709 | 0.7105 | 0.6372 |
| [72] | Meditron | 1024 | 0.7595 | 0.7293 | 0.7162 | 0.8085 |
| [72] | Meditron | 2048 | 0.7599 | 0.7225 | 0.7003 | 0.8306 |
| [70] | Gatortron-Large | 512 | 0.7863 | 0.7751 | 0.7900 | 0.7827 |
| [70] | GatorTron-Medium | 512 | **0.7921** | **0.7653** | **0.7451** | **0.8453** |
| [64] | BiomedBERT-abstract | 512 | 0.7741 | 0.7517 | 0.7457 | 0.8048 |
| [64] | BiomedBERT-full | 512 | 0.7589 | 0.7303 | 0.7195 | 0.8029 |
| [73] | Clinical BERT | 512 | 0.7565 | 0.7361 | 0.7386 | 0.7753 |
| [65] | Bio ClinicalBERT | 512 | 0.7502 | 0.7167 | 0.7026 | 0.8048 |
| [65] | Bio Discharge Summary BERT | 512 | 0.7492 | 0.7040 | 0.6786 | 0.8361 |
| [66] | RoBERTa-Large | 512 | **0.7984** | **0.7595** | **0.7170** | **0.9006** |
| [66] | RoBERTa-Base | 512 | 0.7789 | 0.7342 | 0.6951 | 0.8858 |
| [67] | CORe-clinical | 512 | 0.7476 | 0.7205 | 0.7155 | 0.7827 |
| [70] | Gatortron-Base | 512 | 0.7917 | 0.7663 | 0.7488 | 0.8398 |
| [69] | Clinical-Longformer | 512 | 0.7707 | 0.7498 | 0.7474 | 0.7956 |
| [69] | Clinical-Longformer | 1024 | 0.7949 | 0.7653 | 0.7389 | 0.8600 |
| [69] | Clinical-Longformer | 2048 | 0.7897 | 0.7624 | 0.7423 | 0.8435 |
| [68] | Clinical-Bird | 512 | 0.7772 | 0.7303 | 0.6900 | 0.8895 |
| [68] | Clinical-Bird | 1024 | 0.7638 | 0.7332 | 0.7180 | 0.8158 |
| [68] | Clinical-Bird | 2048 | 0.7795 | 0.7449 | 0.7178 | 0.8527 |

specifically 2048, achieved the best performance with F1 score, accuracy, precision, and recall scores of 0.7415, 0.7176, 0.7185, and 0.7661, respectively. Asclepius also achieved comparable results, with an F1-score of 0.76, using the 512 and 1024 sequence lengths. Three models from Asclepius-R, each with sequence lengths of 512, 1024, and 2048, were fine-tuned. Notably, the 512 sequence length model achieved the best outcomes with scores of 0.7645 for F1-score, 0.7186 for accuracy, 0.6857 for precision, and 0.8637 for recall. The Meditron model exhibited similar performance, where the 2048 sequence length model achieved the best outcomes. Regarding GatorTron, the medium size model (i.e., GatorTron-Medium, with a sequence length of 512) achieves better results than GatorTron-Large, which maintained the same sequence length, yielding scores as follows: F1 score (0.7921), accuracy (0.7653), precision (0.7451), and recall (0.8453). The superior performance of GatorTron-Medium may be attributed to the size of the fine-tuning dataset, which is more appropriately scaled for optimizing the medium model than the large one. When evaluating all LLM models, GatorTron-Medium consistently achieved the best results with a sequence length of 512 ($P < 0.05$). Various LM models of different sizes have been fine-tuned and tested, with the test results presented in Table 4. BiomedBERT Abstract and Full were tested; BiomedBERT Abstract with a sequence length of 512 produced superior results (7741, 0.7517, 0.7457, and 0.8048 for F1 score, accuracy, precision, and recall, respectively). ClinicalBERT model demonstrated better performance compared to its extended version, BioClinicalBERT, achieving 0.7565, 0.7361, 0.7386, and 0.7753 for F1-score, accuracy, precision, and recall, respectively, using a 512 sequence length. In contrast, Bio-Discharge-Summary-BERT with a 512 sequence length

underperformed compared to other LM models despite being fine-tuned with discharge summaries. With a sequence length of 512, RoBERTa-Large and RoBERTa-Base were fine-tuned and tested. The test results indicate that RoBERTa-Large outperforms the base version ($P < 0.05$), achieving 0.7984, 0.7595, 0.7170, and 0.9006 for F1-score, accuracy, precision, and recall, respectively. The Clinical-Longformer model was fine-tuned with three sequence lengths (512, 1024, and 2048). The model using a 1024 sequence length exhibited the best performance among the three, with outcomes of 0.7949, 0.7653, 0.7389, and 0.8600 for F1-score, accuracy, precision, and recall, respectively. The 2048-based model also showed better results than the 512-based model. Similarly, the Clinical-Bird model was fine-tuned with three sequence lengths. Here, the model with the longest sequence length achieved the best results compared to the others (0.7795, 0.7449, 0.7178, and 0.8527 for F1-score, accuracy, precision, and recall, respectively). Two other models, CORe-clinical and Gatortron-Base, were examined using a 512 sequence length. CORe-clinical had the lowest scores among all LMs (0.7476, 0.7205, 0.7155, and 0.7827 for F1-score, accuracy, precision, and recall, respectively), whereas Gatortron-Base showed superior performance (0.7917, 0.7663, 0.7488, 0.8398 for F1-score, accuracy, precision, and recall, respectively), comparable to the best LM model, Clinical-Longformer. The top-performing LLM is GatorTron-Medium, and the best LM is Clinical-Longformer. Although LLM models generally perform better than LM models on average, no significant difference in performance was observed between the best LLM and the best LM. In addition to the comparative performance evaluation of LLM and LM on radiology reports, the experiments of this Section and the previous Section show how discharge summaries and radiology reports contribute differently to predicting ICU mortality in mentally disordered patients. The combination of the two modalities is expected to improve the performance of the language models. There are two techniques for modality fusion, i.e., early and late fusion. In the upcoming experiments, we explore the performance of both techniques.

## 4.4 Results based on multimodal data: Early fusion

This experiment investigates the research question RQ4. We explore early fusion by combining the two types of reports (i.e., discharge summaries and radiology reports) and using them as input for the language model. As shown in Table 5, the Clinical-Longformer model achieves the best and is evaluated with two different sequence lengths, 1024 and 2048. For the 1024 sequence length, it attains an F1-Score of 0.7721, while the model with a longer sequence length of 2048 improves its F1-Score to 0.7788. The slightly improved performance with a longer sequence length could indicate that this model benefits from having more input data, especially for capturing complex patterns across both discharge summaries and radiology reports. Despite using multimodal data through early fusion, the results indicate that no significant improvements were achieved over single-mode models. This could be attributed to the limited input sequence length or the complexity of integrating different types of medical reports. This encourages us to investigate late fusion to optimize a multimodal approach in ICU mortality prediction.

## 4.5 Results based on multimodal data: Late fusion with ensemble model

This experiment investigates the research questions RQ5, RQ6, and RQ7. Based on radiology reports and discharge summaries, the results of the LLM and LM models fell within the 70s. These language models exhibit varying strengths and weaknesses due to differences in data sources, architectural designs, and hyperparameters. Consequently, we explore the use

**Table 5. The results of LLM and LM are based on the early fusion of discharge summary and radiology reports.**

| Ref. | Model | Sequence Length | F1-Score | Accuracy | Precision | Recall |
|------|-------|-----------------|----------|----------|-----------|--------|
| [63] | Asclepius | 2048 | 0.7530 | 0.7479 | 0.7380 | 0.7685 |
| [70] | GatorTron-Medium | 512 | **0.7710** | **0.7649** | **0.7517** | **0.7913** |
| [66] | RoBERTa-Large | 512 | 0.7662 | 0.7365 | 0.6886 | 0.8636 |
| [69] | Clinical-Longformer | 1024 | 0.7721 | 0.7345 | 0.6762 | 0.8997 |
| [69] | Clinical-Longformer | 2048 | **0.7788** | **0.7649** | **0.7355** | **0.8274** |

of a voting ensemble to harness the complementary strengths of different models. The resultant ensemble improves robustness, generalization, and accuracy by integrating contributions from each model. Note that ensemble models are effective in reducing biases, errors, and uncertainties inherent in individual models. Furthermore, they produce a medically accepted classifier, as each base model functions as an expert within the ensemble's mixture of experts. In this experiment, we examine the role of multimodality fusion and voting ensemble in improving the classification results. Early-fusion and late-decision fusion strategies are investigated. Initially, we attempt early fusion by combining two types of reports (i.e., discharge summary and radiology report) and inputting them into the language model. We observe that no significant improvements are attained compared to the models based on a single model, which may be attributed to the length of the prompt (See Table 5). A significant performance enhancement is demonstrated for the decision fusion strategy, as shown in Table 6. Different ensemble strategies are explored using various numbers and types of language models with diverse input reports. Empirically, the optimal number of base models is three. Incorporating more than three base models, such as five different LLMs and LMs, does not yield substantial accuracy improvements relative to the required memory and processing resources. Table 6 displays the results of the ensemble classifiers. We explored the influence of heterogeneous base classifiers and heterogeneous input data on the ensemble's performance. The table has three sections: the first section presents the ensemble results based on the discharge summary, the second section details results based on the radiology report, and the third section is dedicated to combined discharge and radiology data results. Each section examines the role of heterogeneous language models in enhancing ensemble accuracy. Note that Table 6 only displays the ensemble models that achieve the best performance. These abbreviations are used in the discussion: A (Asclepius), DS (Discharge Summary), RR (Radiology Reports), CL (Clinical-Longformer), GM (GatorTron-Medium), GB (Gatortron-Base), GL (Gatortron-Large), CB (Clinical-Bird), and RL (RoBERTa-Large). For ensemble models based on DS data, we note improvement in all models (exceeding 81%) compared to the LLMs and LMs based on the same input data. The combination of A, GM, and CL achieves the best outcomes with F1-scores, accuracy, precision, and recall of 0.8190, 0.8037, 0.7597, and 0.8884, respectively. Heterogeneous ensembles based on RR do not surpass the results of LLMs and LMs-based classifiers. Surprisingly, in line with the theory of ensemble modeling, the integration of heterogeneous modalities with heterogeneous base classifiers delivers the most effective results compared to other ensemble configurations. Integrating diverse base classifiers using varied input data enhances the overall ensemble's performance. The combination of A with DS, CL with DS, and CL with RR significantly outperforms other models ($P < 0.05$) and achieves the highest results of 0.8221, 0.8068, 0.7619, and 0.8926 for F1-score, accuracy, precision, and recall, respectively. We observe that the quality of the input data significantly influences the performance of the ensemble, as demonstrated in the results from an ensemble based on two DS data: GM with DS, CL with DS, and CB with RR, which achieved F1-scores, accuracies,

**Table 6. Results of ensemble language models.** Abbreviations: DS: Discharge Summary, RR: Radiology Reports, A: Asclepius, CL: Clinical-Longformer, GM: GatorTron-Medium, GB: Gatortron-Base, GL: Gatortron-Large, CB: Clinical-Bird, RL: RoBERTa-Large.

| Modalities | Models | F1-Score | Accuracy | Precision | Recall |
|---|---|---|---|---|---|
| Discharge Summary | A, GM, and CL | **0.8190** | **0.8037** | **0.7597** | **0.8884** |
| | GM, RL, and CL | 0.8152 | 0.7986 | 0.7531 | 0.8884 |
| | A, RL, and CL | 0.8144 | 0.8027 | 0.7688 | 0.8657 |
| | A, GM, and RL | 0.8140 | 0.7975 | 0.7526 | 0.8864 |
| | GM, CL, and CB | 0.8137 | 0.7975 | 0.7535 | 0.8843 |
| | A, CL, and CB | 0.8136 | 0.8017 | 0.7674 | 0.8657 |
| | A, RL, and CB | 0.8112 | 0.7975 | 0.7599 | 0.8698 |
| | A, GB, and CL | 0.8102 | 0.7934 | 0.7491 | 0.8822 |
| Radiology Report | RL, CB, and CL | **0.7787** | **0.7469** | **0.6918** | **0.8905** |
| | GL, CB, and CL | 0.7756 | 0.7531 | 0.7108 | 0.8533 |
| | GL, GB, and CL | 0.7740 | 0.7593 | 0.7294 | 0.8244 |
| | GB, CB, and CL | 0.7738 | 0.7469 | 0.6995 | 0.8657 |
| | GM, GB, and CL | 0.7733 | 0.7510 | 0.7098 | 0.8492 |
| | GM, CB, and CL | 0.7712 | 0.7438 | 0.6967 | 0.8636 |
| Combined Reports | (DS) A, (DS) CL, and (RR) CL | **0.8221** | **0.8068** | **0.7619** | **0.8926** |
| | (DS) GM, (DS) CL, and (RR) CB | 0.8200 | 0.7996 | 0.7441 | 0.9132 |
| | (DS) GM, (DS) CL, and (RR) GL | 0.8196 | 0.8027 | 0.7548 | 0.8967 |
| | (DS) RL, (DS) CL, and (RR) CL | 0.8178 | 0.8017 | 0.7561 | 0.8905 |
| | (DS) GB, (DS) CL, and (RR) CL | 0.8167 | 0.7955 | 0.7399 | 0.9112 |
| | (DS) CL, (DS) CB, and (RR) CL | 0.8159 | 0.7996 | 0.7544 | 0.8884 |
| | (DS) GM, (DS) CB, and (RR) GB | 0.8152 | 0.7944 | 0.7403 | 0.9070 |
| | (DS) RL, (DS) CL, and (RR) CB | 0.8149 | 0.7996 | 0.7571 | 0.8822 |
| | (DS) A, (DS) CL, and (RR) GB | 0.8146 | 0.8006 | 0.7612 | 0.8760 |

precisions, and recalls of 0.8200, 0.7996, 0.7441, and 0.9132, respectively. Furthermore, LLMs have a more pronounced impact on the performance of the ensemble than LMs, as evidenced by the previous two ensembles. These findings align with those reported in the literature on ensemble models' performance. The results demonstrate that LLMs can utilize unstructured clinical notes to enhance mortality prediction for ICU patients with mental disorders. Additionally, integrating discharge summaries and radiology reports via late fusion significantly improves the accuracy of these predictive models. This improvement comes from the crucial role of ensemble modeling in strengthening the robustness and generalizability of ICU mortality prediction models for patients with mental disorders. In summary, we can clearly notice that LLM can improve ICU mortality prediction for patients with mental disorders by leveraging unstructured clinical notes such as discharge summaries and radiology reports. These notes provide a wealth of information about patients' medical history, treatments, symptoms, and diagnostic results, which LLMs can interpret to identify risk factors associated with mortality. In addition, the fusion of multimodal data, such as the combination of discharge summaries with radiology reports, greatly improves the predictive performance of LLM when integrated through ensemble decision-making models. Multimodal fusion enables the model to process diverse data, capturing a wider range of clinical patterns and complex patient conditions for more robust mortality prediction. The confusion matrices that visualize the model performance in predicting patient mortality (alive vs. dead) are shown in Fig 5. These matrices represent the performance of the best classifiers in Table 6.

Fig 6 summarizes the average performance of all techniques discussed in this study, highlighting the models that perform best. The results demonstrate that late fusion ensemble models with combined discharge summaries and radiology reports (LF-MM-Combined) consistently achieved the highest performance in most metrics, indicating their superior ability

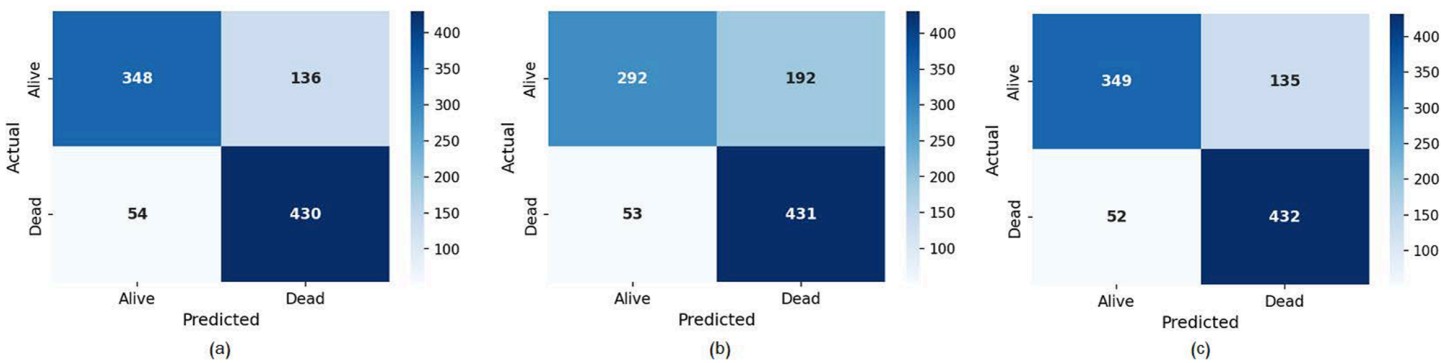

**Fig 5. Confusion matrix of best classifiers in Table 6 by using: (a) Discharge summaries only, (b) radiology reports only, and (c) combined discharge summaries and radiology reports.**

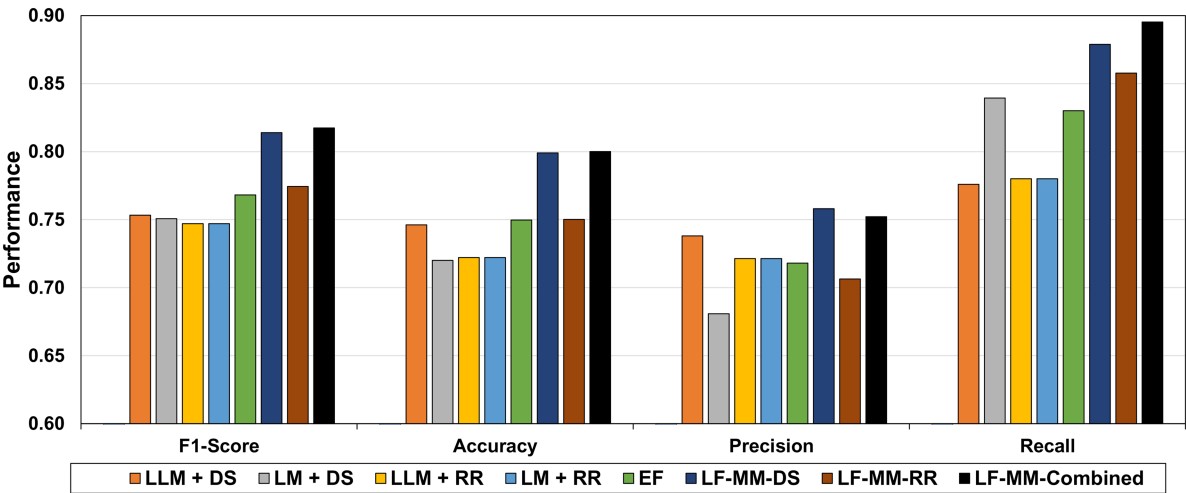

**Fig 6. Comparison of the average performance of all discussed techniques, including single modality LLM/LM models, early and late fusion LLM/LM models.** The evaluated categories include: LLM + DS: Large language models with discharge summaries, LM + DS: Language models with discharge summaries, LLM + RR: Large language models with radiology reports, LM + RR: Language models with radiology reports, EF: Early fusion of DS and RR, LF-MM-DS: Late fusion ensemble models with discharge summaries, LF-MM-RR: Late fusion ensemble models with radiology reports, LF-MM-Combined: Late fusion ensemble models with combined discharge summaries and radiology reports.

to integrate diverse medical data sources. The second best performing model was LF-MM-DS (late fusion ensemble with discharge summaries), followed by LF-MM-RR (late fusion ensemble with radiology reports) as the third best performing model. These findings suggest that ensemble models, especially when combining multiple data sources, improve prediction performance in ICU mortality prediction tasks.

## 4.6 Comparison with non-LLM techniques

Several techniques have been proposed in the literature to analyze and classify non-structural data, including medical notes [79,80]. We have explored the performance of several popular techniques, including convolutional neural network (CNN) [81], LSTM [82], bidirectional long-short-term memory (BiLSTM), and recurrent convolutional neural network (RCNN). Table 7 presents the results of four non-LLM neural network models (LSTM, BiLSTM, CNN,

**Table 7. Results of non-LLM neural network models on mortality prediction using single modalities (i.e., discharge Summaries and radiology reports) and multimodal early fusion.**

| Modalities | Neural Network | F1-Score | Accuracy | Precision | Recall |
|---|---|---|---|---|---|
| Discharge Summary | LSTM | 0.6963 | 0.7084 | 0.7276 | 0.6817 |
| | BiLSTM | 0.7169 | 0.7189 | 0.7234 | 0.7141 |
| | CNN | 0.7129 | 0.7192 | 0.7293 | 0.699 |
| | RCNN | 0.7481 | 0.7521 | 0.7633 | 0.7359 |
| Radiology Report | LSTM | 0.6993 | 0.6869 | 0.7114 | 0.6889 |
| | BiLSTM | 0.7101 | 0.6892 | 0.7038 | 0.7219 |
| | CNN | 0.6807 | 0.6629 | 0.6812 | 0.6893 |
| | RCNN | 0.6991 | 0.6992 | 0.7436 | 0.6738 |
| Early Fusion | LSTM | 0.6858 | 0.6936 | 0.705 | 0.6694 |
| | BiLSTM | 0.707 | 0.7146 | 0.7268 | 0.6911 |
| | CNN | 0.6752 | 0.6901 | 0.7094 | 0.6509 |
| | RCNN | 0.7137 | 0.7263 | 0.7488 | 0.6822 |

and RCNN) evaluated in mortality prediction using three different data configurations: discharge summaries, radiology reports, and early fusion of both. Since these non-LLM models are not trained on medical reports, all reported results are based on 5-fold cross-validation, and the average results have been reported.

Using discharge summaries, the best performing model is RCNN with an F1 score of 0.7481, accuracy of 0.7521, precision of 0.7633, and recall of 0.7359. This model significantly outperforms the others in all metrics, highlighting its ability to capture both sequential and contextual information in the text. For radiology reports, the BiLSTM model leads with an F1 score of 0.7101, indicating its strong performance in capturing information from this modality. It surpasses the RCNN model, which performed better in discharge summaries. The RCNN model follows closely with an F1 score of 0.6991, showing that, while still strong, it does not perform as well on radiology reports as it does on discharge summaries. In the early fusion modality, RCNN returns to the top with the highest F1 score of 0.7137, reaffirming its strength when combining multiple data sources. This suggests that the RCNN model can efficiently utilize both discharge summaries and radiology reports to make predictions. However, compared to the performance of LLM and LM, there is a significant performance gap, with the best-performing models achieving 0.7982 and 0.7852 for LLM and LM, respectively. LM ensemble models that integrate LMs have set a much higher benchmark for classification tasks, particularly in late fusion techniques. In contrast to these non-LLM neural networks, a late fusion ensemble model using LMs achieves a significantly higher F1-score of 0.8221. This marked improvement in performance highlights the ability of LLMs to capture complex textual features and leverage domain-specific knowledge more effectively than traditional models like CNN, LSTM, and BiLSTM. Although RCNN is the best-performing non-LLM model, it still falls short of the LLM ensemble in terms of F1-score. This shows that LLMs are better equipped to handle the intricacies of clinical text and integrate information from multiple sources, resulting in higher precision in predicting mortality in ICU settings.

A key aspect of the advanced performance of LMs/LLMs is their capability to grasp context on a deeper semantic level. In contrast, non-LM techniques rely heavily on syntactic features and often lack the advanced contextual understanding that LMs grasp. Moreover, the use of late fusion in LLM ensemble models enhances their ability to generalize better by combining outputs from multiple models, further boosting the classification performance, outperforming what standalone models can achieve. In summary, while CNN and BiLSTM provide solid results, they fall short of the best-performing LLM ensemble models, which outperform them

by a wide margin, achieving an F1 score of 0.8221. This indicates that modern LMs offer a significant advantage in terms of accuracy and overall model robustness for complex tasks such as medical note classification.

## 5 Limitations and future directions

This study introduces a novel ICU mortality prediction framework based on two text-based modalities for patients with mental disorders that represents a unique contribution to the medical domain, as this is the first study to predict patient mortality solely based on text data. Furthermore, it confirms the established fact that ensemble models comprising diverse base models using heterogeneous data generally outperform individual base classifiers. The study results surpassed those of the previous literature. We explore the significant potential of language models for developing clinical decision support systems in the critical domain of the ICU. However, the study has several limitations that will be addressed in future research. First, since the framework was developed and evaluated solely on internal clinical notes, its generalizability to other healthcare settings, where documentation styles, terminology, and patient populations may differ, remains uncertain. Future work should focus on validating the proposed approach with external datasets from diverse institutions to ensure broader applicability and robustness in other real-world clinical environments. In addition, the dataset was balanced with respect to the binary mortality outcome by matching the number of deceased and surviving patients with mental disorders. However, no stratification was performed on the basis of comorbidity profiles. We acknowledge that the distribution of comorbidities may influence model performance and recommend further stratified analysis in future studies. Secondly, because ICU data inherently are time-series data, we explore methods to make these data comprehensible to language models and allow the models to monitor temporal changes over time. Thirdly, we will investigate the role of multimodal data in varying formats, such as structured and image data, to enhance the decision-making of LLM and LM models. This exploration is crucial since medical experts employ various modalities, such as vital signs and images, to develop reliable and medically significant models. Fourthly, language models of different sizes are inherently black-box. The decisions of these models are not considered trustworthy [44], as domain experts consistently require explanations for specific decision-making processes. We extend the current work by incorporating explainability features into the data. Various XAI techniques that support the robustness of the model decision will be examined, e.g., attention heatmaps, SHAP, and saliency-based methods [83]. Finally, we investigate the critical role of ensemble models in creating stable and accurate classifiers. We will explore various ensembling algorithms, including stacking and dynamic ensembles [84,85]. Additionally, we will examine different techniques to enhance data fusion, such as cross-attention. A detailed analysis of the relationship between ensemble complexity and performance will be conducted. Furthermore, we will investigate the trade-off between model fine-tuning and the application of external knowledge using the Retrieval Augmented Generation (RAG) technique.

## 6 Conclusion

This work introduces an innovative method to predict the risk of mortality in the ICU for patients with mental disorders. The proposed ensemble framework utilizes LMs/LLMs and unstructured clinical notes, including discharge summaries and radiology reports. The results of our study suggest that the use of a combination of advanced language models significantly enhances the accuracy of mortality predictions compared to single LM or LLM models. The ensemble models, which incorporate heterogeneous LMs/LLMs and various text modalities,

demonstrate improved performance. This underscores the importance of using diverse data sources and multiple models to produce robust and stable classifiers. The findings highlight the substantial potential of employing only unstructured medical texts in clinical decision making, potentially leading to more personalized and proactive care approaches. The suggested framework improved prediction accuracy and provided a comprehensive understanding of the patient's health status. In this study, we showcase a pioneering case study on the use of language models and medical notes to construct an accurate clinical decision support system for ICU patients. The results emphasize the critical roles of the fine-tuning step, the integration of multimodal data sources, and the ensemble approach with diverse base classifiers in enhancing the performance of the resultant decision support model. In future work, various approaches will be explored to further improve the accuracy of predictive models. For instance, we will explore combining supplementary medical modalities of other patients, such as structured data and imaging. Models developed using LLM and LM could lead to intelligent systems that significantly enhance medical decision-making and patient quality of life.

## Author contributions

**Conceptualization:** Waleed Nazih.

**Data curation:** Waleed Nazih.

**Formal analysis:** Tamer Abuhmed, Meshal Alharbi, Shaker El-Sappagh.

**Investigation:** Shaker El-Sappagh.

**Methodology:** Waleed Nazih, Tamer Abuhmed, Meshal Alharbi.

**Project administration:** Meshal Alharbi.

**Software:** Waleed Nazih.

**Supervision:** Tamer Abuhmed.

**Validation:** Tamer Abuhmed, Meshal Alharbi, Shaker El-Sappagh.

**Visualization:** Shaker El-Sappagh.

**Writing – original draft:** Waleed Nazih, Shaker El-Sappagh.

**Writing – review & editing:** Tamer Abuhmed, Meshal Alharbi.

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
