## [Decision Letter · Decision Letter 0]

24 Sep 2024

PONE-D-24-32699Mortality Prediction for ICU Patients with Mental Disorders Using Large Language Models Ensemble and Unstructured Medical NotesPLOS ONE

Dear Abuhmed,

Thank you for submitting your manuscript to PLOS ONE. After careful consideration, we feel that it has merit but does not fully meet PLOS ONE’s publication criteria as it currently stands. Therefore, we invite you to submit a revised version of the manuscript that addresses the points raised during the review process. Please submit your revised manuscript by Nov 08 2024 11:59PM. If you will need more time than this to complete your revisions, please reply to this message or contact the journal office at plosone@plos.org. Please include the following items when submitting your revised manuscript:

We look forward to receiving your revised manuscript.

Kind regards,

Menglin Yang

Academic Editor

PLOS ONE

“The authors extend their appreciation to the King Salman Center for Disability 687 Research for funding this work through Research Group no KSRG-2023-101.”

“King Salman Center for Disability Research for funding this work through Research Group no KSRG-2023-101”

5. Please note that your Data Availability Statement is currently missing [the repository name]. If your manuscript is accepted for publication, you will be asked to provide these details on a very short timeline. We therefore suggest that you provide this information now, though we will not hold up the peer review process if you are unable.

Reviewers' comments:

Reviewer's Responses to Questions

**Comments to the Author**

1. Is the manuscript technically sound, and do the data support the conclusions?

Reviewer #1: Yes

Reviewer #2: Yes

Reviewer #3: Yes

2. Has the statistical analysis been performed appropriately and rigorously? 

Reviewer #1: Yes

Reviewer #2: No

Reviewer #3: I Don't Know

3. Have the authors made all data underlying the findings in their manuscript fully available?

Reviewer #1: Yes

Reviewer #2: No

Reviewer #3: No

4. Is the manuscript presented in an intelligible fashion and written in standard English?

Reviewer #1: Yes

Reviewer #2: Yes

Reviewer #3: Yes

5. Review Comments to the Author

Reviewer #1: This paper proposes an ensemble framework using large language models (LLMs) and language models (LMs) to predict mortality risk for ICU patients with mental disorders, based on unstructured clinical notes. The study explores different model architectures, input modalities, and ensemble strategies to improve prediction accuracy.

Statistical analyses:

1- Standard classification metrics (accuracy, precision, recall, F1) are used.

2- Results are presented comprehensively across different models and configurations.

3- Statistical significance tests are not reported, which would strengthen comparisons.

4- Confidence intervals or error bars on results would help assess reliability.

Data and Analysis Evaluation:

Use of MIMIC-IV provides a large, diverse dataset

Appropriate filtering to focus on patients with mental disorders

Balanced dataset creation helps avoid bias

Inclusion of both discharge summaries and radiology reports is valuable

The sample size of 11,914 patients is substantial. Data preprocessing steps are described, though more details on handling of missing data would be helpful.

Writing Quality Check:

The paper is generally well-written and clearly organized. The methodology and results are presented in a logical flow.

# Minor comments that should be resolved or included:

1- Abstract could be more concise and highlight key findings more clearly

2- Some sentences are overly long and complex, impacting readability

3- A few grammatical errors and typos need correction

4- Figures could be improved for clarity, especially Fig. 1

5- Lack of comparison to non-LLM baselines or clinical risk scores, should be included

6- No ablation studies to isolate impact of different components, should be included

7- Lack of complexity analysis of Fine-tuning LLMs, should be included

8- Limited exploration of early fusion or other fusion strategies, should be included

9- Missing statistical significance tests for model comparisons

10- Inadequate discussion of limitations and potential biases

Reviewer #2: This study is an interesting study on predicting mortality rates in ICU patients with mental disorders from unstructured medical notes using a LLM ensemble method. While the idea for the study is very relevant in the contemporary times, there are some improvements that this study needs to make it suitable for publication:

1. The paper would benefit from condensing the points of focus into specific Research Questions (RQ), which can then be referred to in the discussions later in the paper to connect the relevant findings with specific RQs for a more complete discussion of the results.

2. F1 score is the metric for evaluation used in this paper, which might be a common method in this usecase but the authors still needed to substantiate with appropriate citations for their reasoning to use it.

3. The paragraphs need to have a 'justified' alignment.

4. The tables need to be more legible.

5. The results could be better presented with some form of visualisation.

Reviewer #3: Title: Mortality Prediction for ICU Patients With Mental Disorders Using Large Language Models Ensemble and Unstructured Medical Notes

Summary

The article presents the comparison of various LM/LLM model configurations and tuning for predicting in-hospital (ICU) deaths of patients determined to be suffering from mental health disorder. The study uses four LLM and four LM with textual data derived from the MIMIC IV dataset (discharge summaries and radiology reports). Results are presented for a balanced (equal number of patients with/without mental health disorder)

Strengths

The article is well written and easy to read.

The authors do a good job in motivating the paper, i.e. mental health as a risk factor for mortality in ICU admissions.

The literature review is thorough and clearly introduces past efforts and state-of-the-art ML/DM techniques for utilising text-based input to predict in-hospital mortality.

The LM/LLMs selected for the study are good.

The evaluation of the various forms of the selected LM/LLM - out of the box, tuned, ensemble and different modalities for tuning is interesting and informative.

The paper's method and findings do address the research questions raised in the Introduction.

Major Weaknesses

It is not clear how the patients were assigned to "mental disorder". The authors mention ICD-10 code beginning with "F" relating to mental health disorders. However, it is not clear how the "F" ICD-10 code is used. Is a single "F" code in the patient's entire history sufficient to set this flag in the input feature? Is the feature binary (has/has not mental health disorder)? Is the specific ICD-10 code included as a feature.

The authors did not make clear the patient distribution in MIMIC IV such as how many of the patients suffered from mental disorders and how many did not, or how many died in ICU and how many survived (for at least one year). This is particularly important in choosing the data - were all patients who died in ICU included and a random selection of the remaining patients included to balance the study dataset then chosen? The study could have been made more robust by choosing multiple partitions of died/not died patients.

I would like to have seen a more detailed description of the dataset used for the study. As far as I can tell, the input consists of Discharge Summary, Radiology, a flag for mental disorder and a target variable (died in ICU).

Description of the training/validation/testing regime is limited. The authors mention the dataset was partitioned 90% training/validation and 10% testing. However, there is no description of the number or repetitions of the experiments, whether different partitioning was used in different experiment runs, etc.

The results reported are the "best" for each model and experiment. It would be useful to know the range of results and whether the "best" is significantly different from the usual or the worst result.

The paper addresses the question of how various LM/LLM models compare in predicting death in ICU. There are already clinical multi-morbidity models in use that generate mortality risk predictions (over various time horizons - 90 day, 1 year, 5 year). These models are already accepted and in clinical use. I think the paper would be strengthened by comparing with one or more of these models.

Concerns

The authors have relied heavily on pre-prints or unpublished archive references (12 out of 67 references). I think this is not a good idea.

To make it easier for the reader to pick out the best performing models, the table rows containing these should be bolded/highlighted.

As far as I can see, the datsets used for training are not made available to the reader.

Corrections

Introduction

anorexia Nervosa -> anorexia nervosa

Section 3.2

(1st sentence) particularly those that involves -> particularly those that involve

Section 3.3

Figure 1 illustr2ates -> illustrates

References

There are many malformed (missing key information such as publisher, etc). These include

There are many malformed references for URLs (author attribution is incorrect). These include [2], [3], [4], [5], [6]

Some references use full journal name, others abbreviations. Referencing should be consistent - suggest using teh full journal name throughout - see [19], [56], [58]

There is inconsistent capitalisation of journal names. Some capitalise each word, others only the first word. Again, the authors are advised to be consistent and to adopt the standard of the journal to which the article has been submitted.

Similarly, there is inconsistent capitalisation in article titles. Some Some capitalise each word, others only the first word. In some, the first word after a colon is capitalised, while in others it is not. Also, there are instances where capitals should be used to refer to an artefact - e.g MIMIC IV rather than Mimic iv (ref [54]) and Mental-LLM rather than Mental llm (ref [26]). Again, the authors are advised to be consistent and to adopt the standard of the journal to which the article has been submitted.

6. PLOS authors have the option to publish the peer review history of their article (what does this mean?). If published, this will include your full peer review and any attached files.

Reviewer #1: No

Reviewer #2: No

Reviewer #3: **Yes: **Robert Andrews

---

## [Author Response · Author response to Decision Letter 1]

26 Oct 2024

Dear Editor and Reviewer,

We thank you for their valuable comments that significantly helped us to enrich the quality of the manuscript. We've carefully considered the reviewers' comments in our revision. We have tried our best to incorporate the reviewers' and editors' suggestions into the revised version of the manuscript.

To address each reviewer's comments carefully, we responded to them point by point in an attached response file with the revised submission. All changes have been highlighted in a track changes copy of the manuscript using the red color.

Thank you.

---

## [Decision Letter · Decision Letter 1]

2 Jan 2025

PONE-D-24-32699R1Mortality Prediction for ICU Patients with Mental Disorders Using Large Language Models Ensemble and Unstructured Medical NotesPLOS ONE

Dear Authors:

Thank you for submitting your manuscript to PLOS ONE. After careful consideration, we feel that it has merit but does not fully meet PLOS ONE’s publication criteria as it currently stands. Therefore, we invite you to submit a revised version of the manuscript that addresses the points raised during the review process.

We look forward to receiving your revised manuscript.

Kind regards,

Menglin Yang

Academic Editor

PLOS ONE

Journal Requirements:

Reviewers' comments:

Reviewer's Responses to Questions

**Comments to the Author**

1. If the authors have adequately addressed your comments raised in a previous round of review and you feel that this manuscript is now acceptable for publication, you may indicate that here to bypass the “Comments to the Author” section, enter your conflict of interest statement in the “Confidential to Editor” section, and submit your "Accept" recommendation.

Reviewer #3: (No Response)

Reviewer #4: All comments have been addressed

2. Is the manuscript technically sound, and do the data support the conclusions?

Reviewer #3: Partly

Reviewer #4: Yes

3. Has the statistical analysis been performed appropriately and rigorously? 

Reviewer #3: Yes

Reviewer #4: Yes

4. Have the authors made all data underlying the findings in their manuscript fully available?

Reviewer #3: No

Reviewer #4: Yes

5. Is the manuscript presented in an intelligible fashion and written in standard English?

Reviewer #3: Yes

Reviewer #4: Yes

6. Review Comments to the Author

Reviewer #3: Title: Mortality Prediction for ICU Patients with Mental Disorders Using Large Language Models Ensemble and Unstructured Medical Notes

Comments:

I thank the authors for addressing my comments. I am happy with the responses for all but three of my comments. All have to do with data.

Firstly, I am still unsure about the inclusion criteria for patients. The authors clearly state the set of ICD codes that identify mental disorders. What I am missing is if inclusion was based on the presence of one or more of these codes in the latest admission, or whether the inclusion is based on the occurrence of one of these codes in the entire patient history. For instance, if a patient has more than one admittance in the database and an early admittance includes an F50-F59 code, but the latest admittance does not show this code, would the patient be included in the study?

Also, is the mental disorder required to be the primary diagnosis or just included as one of the diagnosis codes? In my experience, hospital records may include up to 20 diagnosis codes.

Next, I am still unsure about the structure feature vectors. My understanding is that the only attributes presented include either/both discharge/radiology report and a target, with no other demographic or clinical history information being included. I think this needs clarifying.

Lastly, I feel that not taking into account comorbidities may introduce some bias into the data. What fraction of the patient cohort had potentially life threatening comorbidities (as well as mental disorder)? That is, are mental disorders independent of other conditions? Were comorbidities equally distributed across the [died |did not die] partitions?

I understand that this is not a clinical paper, it is about experimenting with the capabilities of medically trained LLM/LM models. Nevertheless, it is important to understand and acknowledge the characteristics of the data as they impact (or do not impact) on prediction.

I commend the authors on the modifications they have made to the paper. It is indeed improved and will be a significant contribution.

Minor

Spelling mistake: Section 4.4, 1st paragraph - archives -> achieves

Reviewer #4: The article has successfully met all the necessary requirements for publication, adhering to the stipulated guidelines and standards. After thoroughly reviewing its content, structure, and alignment with the publication's objectives, I am pleased to confirm that there are no outstanding issues or points of concern. As such, I have no further questions or comments to raise at this stage.

7. PLOS authors have the option to publish the peer review history of their article (what does this mean?). If published, this will include your full peer review and any attached files.

Reviewer #3: **Yes: **Robert Andrews

Reviewer #4: No

---

## [Author Response · Author response to Decision Letter 2]

13 Jan 2025

Responses to the reviewers' comments are attached as a separate file in the submission.

---

## [Editor Report · Decision Letter 2]

6 Feb 2025

PONE-D-24-32699R2

Mortality Prediction for ICU Patients with Mental Disorders Using Large Language Models Ensemble and Unstructured Medical Notes

PLOS ONE

Dear Dr. Abuhmed,

Thank you for submitting your manuscript to PLOS ONE. After careful consideration, we have decided that your manuscript does not meet our criteria for publication and must therefore be rejected.

Specifically:

**ACADEMIC EDITOR: Please insert comments here and delete this placeholder text when finished.** Be sure to:

Clearly explain how the manuscript fails to meet the PLOS ONE publication criteria. Please ensure your decision is not justified on novelty or impact.Provide specific feedback from your evaluation of the manuscript

For Lab, Study and Registered Report Protocols: These article types are not expected to include results but may include pilot data. Please do not reject for lack of results.

I am sorry that we cannot be more positive on this occasion, but hope that you appreciate the reasons for this decision. Please try to solve reviewer's concerns and resubmit.

Kind regards,

Academic Editor

PLOS ONE

---

## [Author Response · Author response to Decision Letter 3]

11 Mar 2025

Dear Editor,

Thank you for your feedback.

We have included the responses to the reviewers comments in the attached files.

---

## [Decision Letter · Decision Letter 3]

28 May 2025

PONE-D-24-32699R3Mortality Prediction for ICU Patients with Mental Disorders Using Large Language Models Ensemble and Unstructured Medical NotesPLOS ONE

Dear Dr. Abhumed,

Thank you for submitting your manuscript to PLOS ONE. After consideration, we feel that it has merit but does not fully meet PLOS ONE’s publication criteria as it currently stands. Therefore, we invite you to submit a minor revised version of the manuscript that addresses the points raised during the review process.

We look forward to receiving your revised manuscript.

Kind regards,

Minh Huu Nhat Le, MD

Academic Editor

PLOS ONE

Journal Requirements:

1. Please note that your Data Availability Statement is currently missing [the repository name]. If your manuscript is accepted for publication, you will be asked to provide these details on a very short timeline. We therefore suggest that you provide this information now, though we will not hold up the peer review process if you are unable.

Please update your Data Availability statement in the submission form accordingly."

3. Please upload a copy of Figure 1 & 3 which you refer to in your text on your paper. Or if the figure is no longer to be included as part of the submission please remove all reference to it within the text.

4. Please ensure that you refer to Table 1 in your text as, if accepted, production will need this reference to link the reader to the Table.

Additional Editor Comments (if provided):

Reviewers' comments:

Reviewer's Responses to Questions

**Comments to the Author**

1. If the authors have adequately addressed your comments raised in a previous round of review and you feel that this manuscript is now acceptable for publication, you may indicate that here to bypass the “Comments to the Author” section, enter your conflict of interest statement in the “Confidential to Editor” section, and submit your "Accept" recommendation.

Reviewer #5: (No Response)

Reviewer #6: All comments have been addressed

Reviewer #7: All comments have been addressed

2. Is the manuscript technically sound, and do the data support the conclusions?

Reviewer #5: Yes

Reviewer #6: Partly

Reviewer #7: Yes

3. Has the statistical analysis been performed appropriately and rigorously? 

Reviewer #5: Yes

Reviewer #6: N/A

Reviewer #7: Yes

4. Have the authors made all data underlying the findings in their manuscript fully available?

Reviewer #5: Yes

Reviewer #6: No

Reviewer #7: Yes

5. Is the manuscript presented in an intelligible fashion and written in standard English?

Reviewer #5: Yes

Reviewer #6: Yes

Reviewer #7: Yes

6. Review Comments to the Author

Reviewer #5: Paper is well-written, clear and easy-to-follow. The technical soundness and novelties are strong. The experiments are extensive that show all aspects to assess the effectiveness of methods. Fig 2 quality should be increased. I believe this work is sufficient to be published.

Reviewer #6: The idea is okay, but it mostly just applies existing tools (LLMs, ensembling) without doing anything really new.

The paper doesn’t compare the models to basic baselines like logistic regression or ICU scoring systems (e.g., SOFA, SAPS), which makes it hard to judge if these LLMs are actually better.

They use majority voting for the ensemble, but don’t explain why that method was picked or if other fusion methods were tested.

It’s unclear how long clinical notes (which can be huge) were handled. Were they cut off? Summarized? This could affect the results a lot.

Only one dataset (MIMIC-IV) is used, and it’s just for patients with mental disorders. It’s hard to know if the results would work for other types of patients or in different hospitals.

The results don’t include confidence intervals or show if the improvements are statistically solid. Some scores are very close — we don’t know if the gains are real.

The paper doesn’t explain what the models are learning or how decisions are made — there’s no interpretability, which is important in clinical work.

Writing is a bit wordy in places and technical details (like preprocessing or how inputs were built) could be clearer.

Reviewer #7: Overall: This manuscript presents a novel approach to predicting ICU mortality in patients with mental disorders using large language models and unstructured clinical notes, demonstrating considerable contributions with impressive performance results. Although there are several grammatical and typographical errors, they are not significant. However, there are some parts in both methods and experiments that need to be revised.

Strength:

- The introduction provides a comprehensive background on mental health disorders with current statistics from reputable sources (WHO, NIH). It also thoroughly reviews existing approaches for ICU mortality prediction. Based on these bases, this study effectively establishes the novelty of using LLMs for mortality prediction in mentally disordered ICU patients.

- Research questions (RQ1-RQ7) are specific and directly address identified gaps in the literature. The contributions are clearly articulated and aligned with the research objectives.

- The dataset description is presented well, providing clear information about the MIMIC-IV clinical notes and inclusion criteria for mental disorder patients.

- Although the methodology section lacks some foundations to make it more rigorous, it is generally good and can be considered one of the most important contributions of the paper due to its novelty.

- The experimental setup is fair, and the experimental results are impressive.

Weakness:

- The introduction is excessively detailed in certain areas, along with problems in paragraph division (particularly in paragraphs 3 & 4, lines 67-101), creating a disjointed narrative. As a results, research gaps are scattered throughout rather than cohesively presented.

- Despite the adequate general motivation, it is not as compelling as it could be due to the fragmented presentation of research gaps. Another reason is that the connection between mental disorders and ICU mortality risk could be more explicitly linked to the proposed technical approach.

- There are many research questions (RQ1-RQ7). It would be better if these questions can be clustered into logical categories (e.g., "Model Architecture Questions," "Data Modality Questions").

- The statement "The ensemble of heterogeneous language models (LMs) and LLMs effectively addresses these limitations" in the abstract is imprecise, as LLMs are inherently a subset of language models.

- The Methods section lacks a sufficient explanation for using Low-Rank Adaptation (LoRA) over alternative fine-tuning approaches. Figure 2 appears to be directly reused from the original LoRA paper rather than being redrawn specifically for this study's context.

- Data preprocessing steps are minimally described, with limited details on tokenization and handling of medical terminology.

- The claim that "the utilized dataset is balanced with respect to the presence of severe comorbidities" (lines 328-332) lacks supporting evidence such as statistical tests or distribution visualizations.

- Several instances of redundant terminology appear throughout the manuscript (e.g., "LLM models" where"LLM" already means "Large Language Model"), and Section 3.2 contains a duplicated colon at the end of paragraph 2 (line 368).

- Table 2 is too small, making it difficult to verify its contents.

- Some sentences have awkward phrasing or grammatical issues (e.g., lines 512-513, 601, 652-654).

Concern:

- The manuscript lacks visualizations of data distribution and experiments such as mental disorder categories, length distributions of clinical notes, model performance across different patient subgroups, or confusion matrices for the best-performing models.

- The preprocessing for datasets is not described to ensure reproducibility and justification.

- Is the 1B parameter threshold for distinguishing between LMs and LLMs arbitrary or based on evidence?

- Currently, the number of datasets which are employed is not enough, making it difficult to confirm the robustness and generalizability of the proposed method's effectiveness. There is a need to consider expanding the experimental scope in terms of data.

Final decision: Major Revision

7. PLOS authors have the option to publish the peer review history of their article (what does this mean?). If published, this will include your full peer review and any attached files.

Reviewer #5: No

Reviewer #6: No

Reviewer #7: **Yes: **Minh-Toan Dinh

---

## [Author Response · Author response to Decision Letter 4]

9 Jun 2025

Dear Editor and Reviewers,

Thank you for accepting our manuscript with minor revision and for your insightful comments. We greatly appreciate the time and effort you devoted to evaluating our work, your feedback has notably improved the manuscript. We have carefully considered each comment and incorporated your suggestions into the revised version.

Please find a detailed, point-by-point response in the attached document to the submission.

We hope the updated manuscript now fully meets your expectations. Thank you again for your guidance.

---

## [Decision Letter · Decision Letter 4]

4 Aug 2025

PONE-D-24-32699R4Mortality Prediction for ICU Patients with Mental Disorders Using Large Language Models Ensemble and Unstructured Medical NotesPLOS ONE

Dear Dr. Abuhmed,

Thank you for submitting your manuscript to PLOS ONE. I am pleased to inform you that your manuscript has been **provisionally accepted**, pending **minor revisions**. Therefore, we invite you to submit a revised version of the manuscript that addresses the points raised during the review process.

Please revise your manuscript carefully to address all minor remaining issues. In particular, we request that you conduct a thorough check for grammar, spelling, and clarity to ensure the manuscript meets the standards of written academic English. Additionally, please confirm that your manuscript has been screened for any unintentional plagiarism and that all sources are properly cited. While the reviewers have recommended acceptance, we ask that you review and incorporate all minor comments raised throughout the review process. Once these revisions are completed, we will proceed with final acceptance and move the manuscript into production.

We look forward to receiving your revised manuscript.

Kind regards,

Minh Le, MD, PhD candidate

Academic Editor

PLOS ONE

Journal Requirements:

Reviewers' comments:

Reviewer's Responses to Questions

**Comments to the Author**

1. If the authors have adequately addressed your comments raised in a previous round of review and you feel that this manuscript is now acceptable for publication, you may indicate that here to bypass the “Comments to the Author” section, enter your conflict of interest statement in the “Confidential to Editor” section, and submit your "Accept" recommendation.

Reviewer #5: All comments have been addressed

Reviewer #8: All comments have been addressed

Reviewer #9: All comments have been addressed

Reviewer #10: All comments have been addressed

Reviewer #11: All comments have been addressed

2. Is the manuscript technically sound, and do the data support the conclusions?

Reviewer #5: Partly

Reviewer #8: Yes

Reviewer #9: Yes

Reviewer #10: Yes

Reviewer #11: Yes

3. Has the statistical analysis been performed appropriately and rigorously? 

Reviewer #5: Yes

Reviewer #8: Yes

Reviewer #9: Yes

Reviewer #10: Yes

Reviewer #11: Yes

4. Have the authors made all data underlying the findings in their manuscript fully available?

Reviewer #5: Yes

Reviewer #8: Yes

Reviewer #9: Yes

Reviewer #10: Yes

Reviewer #11: Yes

5. Is the manuscript presented in an intelligible fashion and written in standard English?

Reviewer #5: Yes

Reviewer #8: Yes

Reviewer #9: No

Reviewer #10: Yes

Reviewer #11: Yes

6. Review Comments to the Author

Reviewer #5: Latest revision is well-written. Idea is clear. Experimental quantitative results are sufficient. I believe this work is ready for publication.

Reviewer #8: The paper can be accepted for publication.

The paper can be accepted for publication.

The paper can be accepted for publication.

Reviewer #9: I appreciate the authors’ thoughtful and well-structured responses to the initial comments. Most concerns have been addressed with appropriate revisions and clarifications.

Reviewer #10: Authors have addressed all the raised concerns by the reviewers. The quality of the paper is now in a standard scale and reay to be published.

Reviewer #11: In summary, the authors have rigorously addressed previous reviewer concerns and the work represents the solid scientific contribution with clear clinical relevance. They had conducted extensive ablation studies and justified theories to demonstrate their hypotheses. Therefore, I recommend an acceptance for this manuscript.

7. PLOS authors have the option to publish the peer review history of their article (what does this mean?). If published, this will include your full peer review and any attached files.

Reviewer #5: No

Reviewer #8: No

Reviewer #9: No

Reviewer #10: No

Reviewer #11: No

---

## [Author Response · Author response to Decision Letter 5]

12 Aug 2025

The manuscript's last decision is "accepted with revision", and we have attached our response file to this submission.

---

## [Editor Report · Decision Letter 5]

27 Aug 2025

Mortality Prediction for ICU Patients with Mental Disorders Using Large Language Models Ensemble and Unstructured Medical Notes

PONE-D-24-32699R5

Dear Dr. Abuhmed,

We’re pleased to inform you that your manuscript has been judged scientifically suitable for publication and will be formally accepted for publication once it meets all outstanding technical requirements.

Kind regards,

Minh Le, MD, PhD candidate

Academic Editor

PLOS ONE
---

## [Editor Report · Acceptance letter]

PONE-D-24-32699R5

PLOS ONE

Dear Dr. Abuhmed,

I'm pleased to inform you that your manuscript has been deemed suitable for publication in PLOS ONE. Congratulations! Your manuscript is now being handed over to our production team.

Kind regards,

on behalf of

Dr. Minh Le

Academic Editor

PLOS ONE